# Future sea ice weakening amplifies wind-driven trends in surface stress and Arctic Ocean spin-up

Morven Muilwijk [1] ✉, Tore Hattermann [1,2], Torge Martin[3] & Mats A. Granskog [1]

Arctic sea ice mediates atmosphere-ocean momentum transfer, which drives upper ocean circulation. How Arctic Ocean surface stress and velocity respond to sea ice decline and changing winds under global warming is unclear. Here we show that state-of-the-art climate models consistently predict an increase in future (2015–2100) ocean surface stress in response to increased surface wind speed, declining sea ice area, and a weaker ice pack. While wind speeds increase most during fall (+2.2% per decade), surface stress rises most in winter (+5.1% per decade) being amplified by reduced internal ice stress. This is because, as sea ice concentration decreases in a warming climate, less energy is dissipated by the weaker ice pack, resulting in more momentum transfer to the ocean. The increased momentum transfer accelerates Arctic Ocean surface velocity (+31–47% by 2100), leading to elevated ocean kinetic energy and enhanced vertical mixing. The enhanced surface stress also increases the Beaufort Gyre Ekman convergence and freshwater content, impacting Arctic marine ecosystems and the downstream ocean circulation. The impacts of projected changes are profound, but different and simplified model formulations of atmosphere-ice-ocean momentum transfer introduce considerable uncertainty, highlighting the need for improved coupling in climate models.

In recent decades, the Arctic has undergone rapid changes, warming at more than four times the global rate[1] and experiencing extensive sea ice loss[2,3]. The thinning and shrinking of Arctic sea ice extent[4,5] impacts the mechanical and thermodynamic coupling between the atmosphere and the ocean[6–8], which in turn can affect ocean stratification[9,10] and thereby nutrient availability and ecosystems[11]. Wind generally drives the ocean's surface currents and vertical mixing by exerting stress at the surface. In the polar regions, sea ice mediates this coupling[7], to either amplify or dampen the transfer of momentum depending on the rigidity of the ice pack[12], its compactness (ice concentration)[13], and ice surface and bottom roughness[14,15], as illustrated in Fig. 1a.

Rainville and Woodgate[16] suggested that the continued retreat of sea ice could result in the Arctic Ocean becoming more energetic, and increased turbulent mixing has already been reported ref. 17 and the references therein. Furthermore, a negative feedback mechanism dubbed the ice-ocean stress governor[18], which limits the impact of wind stress on the ocean, will likely be less effective with a thinner and less compact ice cover.

In low sea ice concentration scenarios, sea ice behaves like particles flowing freely on the ocean surface, a state known as free drift. With higher concentrations, the ice pack becomes more compact, restricting flow, and wind energy is partly converted into internal stress[19]. When ice is pushed together hard enough, it eventually breaks

[1]Norwegian Polar Institute, Fram Centre, Tromsø, Norway. [2]Complex Systems Group, Department of Mathematics and Statistics, UiT - The Arctic University of Norway, Tromsø, Norway. [3]GEOMAR Helmholtz Centre for Ocean Research, Kiel, Germany. ✉e-mail: morven.muilwijk@npolar.no

**Fig. 1 | Schematic representation of present-day and future Arctic Ocean atmosphere-ocean momentum exchange. a** The current role of sea ice in mediating the impact of wind stress on the ocean. **b** Future changes in Arctic Ocean momentum exchange described in this paper and their impact on the ocean.

Ocean currents are projected to accelerate due to a combination of (1) wind speeds increase, (2) weakening of the ice pack, and (3) reduced ice coverage. Illustration: NPI/Trine Lise Sviggum Helgerud.

apart and piles up into ridges (sea ice deformation). In this process, kinetic energy is dissipated to internal stress and subsequent ice deformation, which depends on the concentration and thickness of the ice. Consequently, internal stress is a key factor in determining the extent to which wind stress is transferred into ocean surface stress[13].

Climate models indicate that the decline in sea ice area and thickness will persist[20,21], resulting in larger[22] and lengthened open-water periods[23]. Recent findings indicate that irrespective of greenhouse gas emissions scenarios, a seasonally ice-free Arctic could occur already within the next decade[24] and is likely to occur by 2050[21]. Simultaneously, winds are strengthening[25,26], and projections show an increase in cyclone frequency and intensity[27]. However, the combined impact of these factors on future ocean surface stress and circulation in the Arctic Ocean remains unknown.

To assess the future Arctic Ocean surface momentum balance and its seasonality, we analyze the seasonality of the various stress terms in the atmosphere-ice-ocean momentum budget using results from 16

climate models from the latest generation of Coupled Model Inter-comparison Project phase 6 (CMIP6)[28]. One of the CMIP6 models, NorESM2-MM[29], is utilized for visualization of spatial patterns and more detailed analysis as it provides detailed enough output for a more precise breakdown in stress components and ocean mixing, offering a case study of the relevant processes at play. We first show that, under a high greenhouse gas emission scenario (ssp585), ocean stress is projected to increase for the period 2015–2100 (Section **Projected change in ocean stress and wind speed**). We then distinguish the contributions from changes in wind and sea ice conditions to these trends, highlighting the effect of reduced momentum transfer to internal stress within the ice pack (Section **Changing wind vs. changing ice**). Moving forward, we address associated changing seasonal disparities (Section **Changing seasonal disparities**). Further, we scrutinize the consistency of our results across the different models and investigate the role of different parameterizations (Section **Reduced dampening effect**) and ice states (Section **Changing ice concentration, thickness and speed**). Finally, we study the impact of

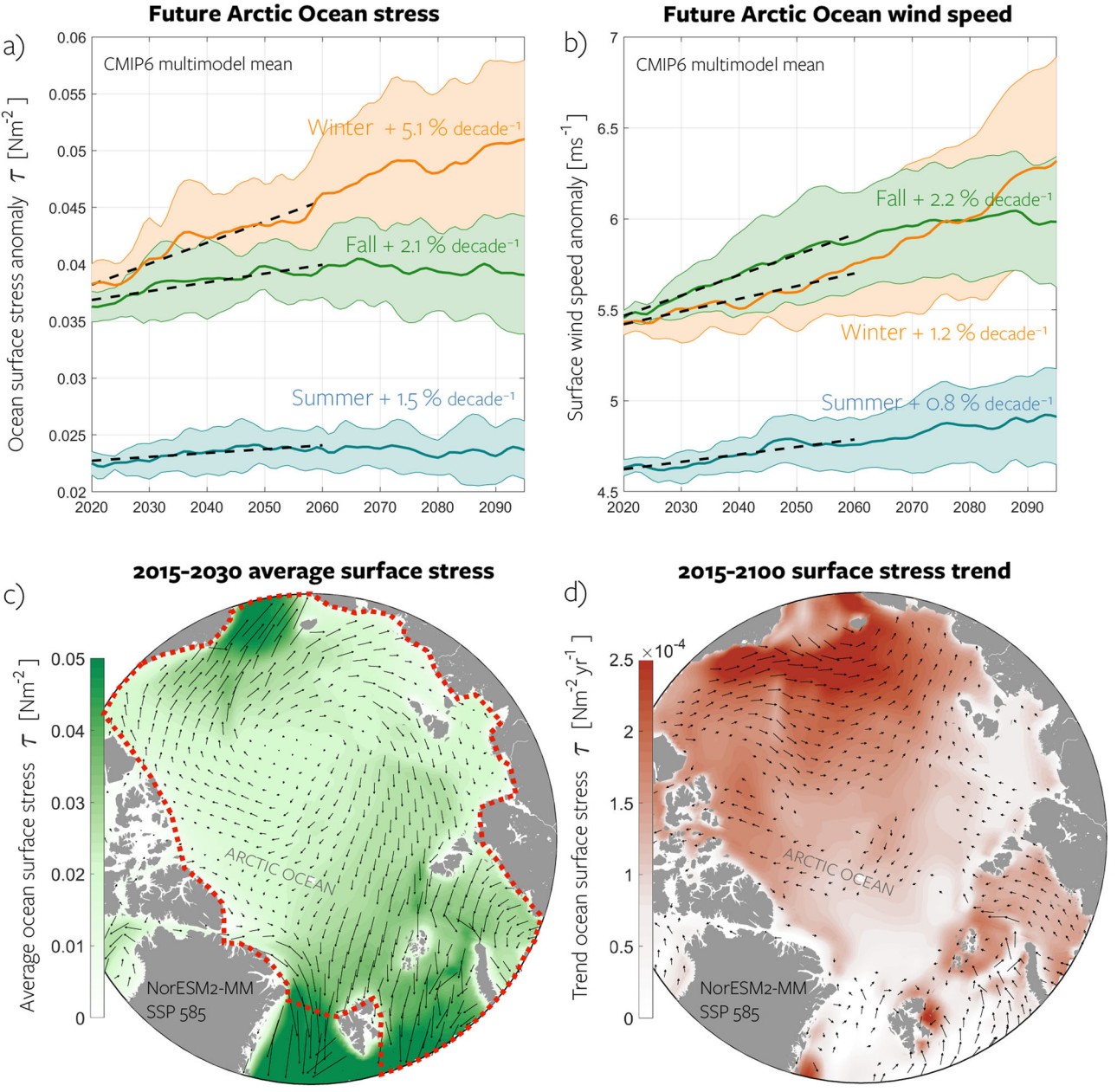

**Fig. 2 | Projections of Arctic ocean stress and wind. a** Multimodel mean Arctic Ocean (region defined by the dashed red line in **c**) surface stress anomalies (relative to 2015–2030 model mean) projected by the CMIP6 models under a high-emission scenario for summer (June–August, blue), fall (September–November, green), and winter (January–March, orange). Envelopes indicate the model spread as determined by one standard deviation. Time series represent a ocean-wide average and have been smoothed using a low-pass filter with a five-year cutoff frequency. The dashed black lines represent the linear trend until 2060, beyond which the stress curves indicate a leveling off. Values indicated in the panel represent the multi-model mean trends over the 2015–2060 period. **b** Same as **a** but for surface wind speed. **c** Spatial representation of the annual mean ocean surface stress for the NorESM2-MM model, serving as an example, during the early part of the century (2015-2030). **d** Linear trend in annual mean ocean surface stress for NorESM2-MM. See the calculation in Methods. Source data are provided as a Source Data file.

changing surface stress on ocean dynamics (Section **Impact on ocean dynamics**).

## Results

### Projected change in ocean stress and wind speed

Multimodel ensemble mean (Methods) ocean surface stress increases across the entire Arctic Ocean and in all seasons (Fig. 2). The Arctic-wide mean increase is statistically significant at the 95% confidence level for fall and winter in 15 out of 16 models (see Supplementary Fig. 1a, b for a comparison of individual model trends). On average, surface stress is lower during summer and higher during fall and winter. The largest trend in surface stress occurs during winter (5.1% per decade), while fall also exhibits a significant trend (2.1% per decade), although this curve flattens after 2060. Figure 2c, d illustrates the spatial pattern of surface stress and projected changes based on the NorESM2-MM model results. Surface stress increases in all regions, with the most prominent trends in the Chukchi Sea and Beaufort Gyre regions, where surface stress is already elevated in the present-day scenario.

Given that surface stress in models typically follows a quadratic drag law[19], predominantly influenced by wind forcing, we can anticipate these trends follow changes in surface wind speed. The CMIP6

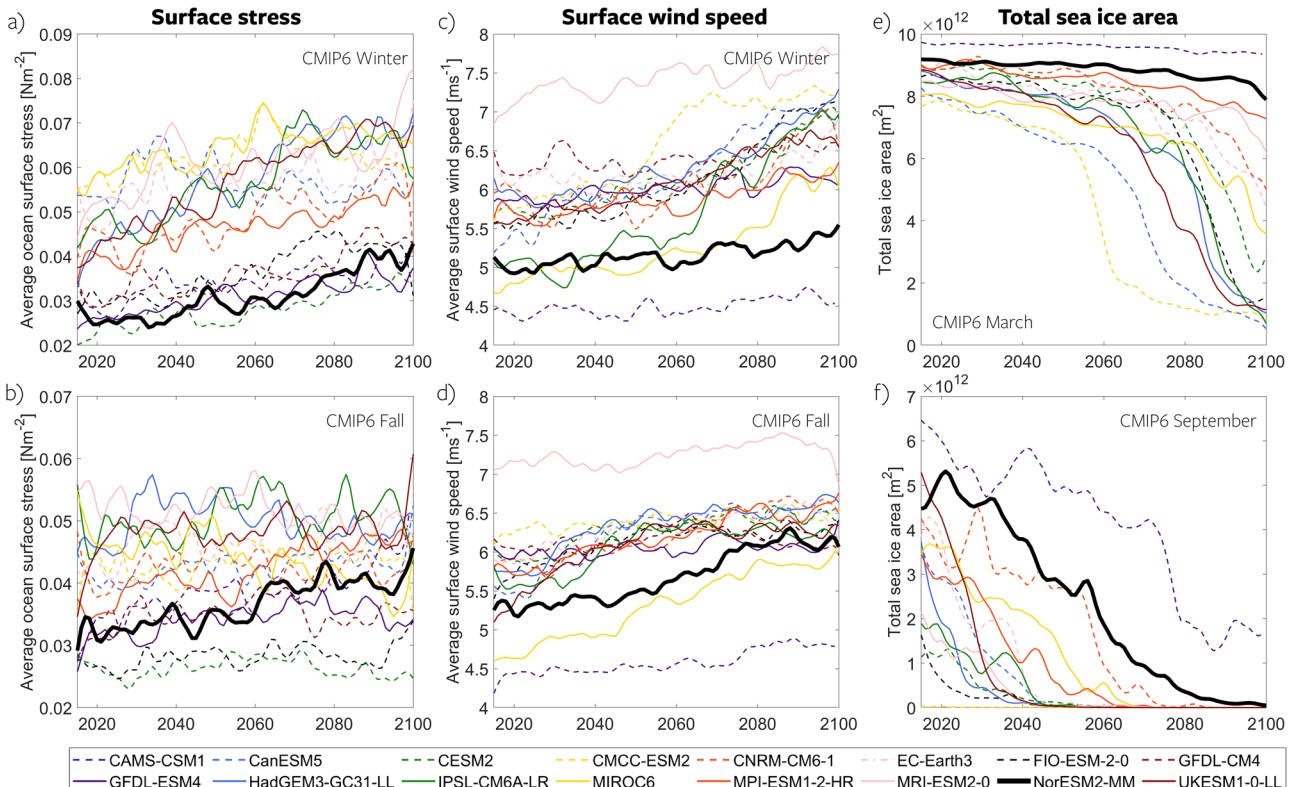

**Fig. 3 | Time series of future Arctic Ocean surface stress, wind speed, and ice area.** Spaghetti plots illustrating model time series from all CMIP6 models, depicting the future change in Arctic ocean surface stress (**a, b**), surface wind speed (**c, d**), and total sea ice area (**e, f**) for winter (top row) and fall (bottom row). The thick black lines represent the NorESM2-MM model, serving as a case study for more detailed analysis. Source data are provided as a Source Data file.

ensemble consistently indicates a strengthening of surface wind speed (Fig. 2b), aligning with previous studies exploring future Arctic wind speeds in earlier CMIP generations[25] and most recently based on CESM2[30]. Similar to surface stress, winds are generally weaker during summer (~4.6 ms⁻¹) in comparison to winter and fall (~5.5 ms⁻¹). However, focusing on the period 2020–2060 (trend lines in Fig. 2a, b), we observe larger trends in wind speed during fall compared to winter (2.2% per decade vs. 1.2% per decade, respectively), while the opposite holds true for surface stress. The surface stress trends are larger in winter than in fall, and this is consistently observed across the entire suite of models (Supplementary Fig. 1c,d). This is surprising given the quadratic law connecting the two quantities. This seasonal discrepancy between changes in wind speed and surface stress implies that the changing sea ice cover plays a crucial role in moderating the transfer of momentum from the wind to the ocean.

Additionally, in summer and fall, wind speed increases linearly through 2100, but surface stress plateaus (Fig. 2a, b). We do not investigate the plateauing in detail but suspect it is related to a change in average sea ice concentration. As discussed in Changing ice concentration, thickness and speed, most models suggest a maximum ocean surface stress for ice concentrations between 10% and 90% (for grid cells with more than 0% ice). Stress increases initially with decreasing ice concentration, but later decreases as basin-mean ice concentration reduces to sub-optimal conditions (as argued by Martin et al.[13]). Consequently, continuously increasing wind speed combined with reduced sea ice concentration can result in a plateau in wind stress. Additionally, the plateauing of the wind stress time series may partly be an artifact of the multi-model averaging, as individual models display a less coherent time evolution (Fig. 3). Further in this paper, we primarily investigate the mechanisms behind the amplified change in stress during winter compared to the changes in wind speed.

Despite consistent trends, the model ensemble shows large differences in mean state for ocean surface stress and surface wind speed (Fig. 3, panels a–d). The average Arctic Ocean surface stress ranges from 0.02 Nm⁻² to 0.055 Nm⁻² across models at the century's start, with this spread remaining constant over time. Similarly, initial wind speeds span from 4.5 ms⁻¹ to 6.9 ms⁻¹ across the model suite, but this spread also remains constant. The spread in stress and wind speed means is not surprising, as both are dependent on the future sea ice state (Fig. 3, panels e-f), wherein models display considerable disparities – a topic that has also been extensively explored in previous studies[31,32]. In the beginning of the century, the total sea ice area in winter is consistent across models. However, the timing and extent of winter sea ice decline differ notably across the individual simulations, and several models sustain a substantial winter sea ice area. In fall, during sea ice minimum, most models exhibit a pronounced decrease in sea ice area, yet certain models demonstrate accelerated ice loss, as corroborated by previous studies[33,34].

Despite significant intermodel disparities in mean state, both wind speed and surface stress demonstrate consistent climate responses across the suite. Notably, no clear correlation emerges between the mean state and the strength of the trends.

In the following, NorESM2-MM (thick black lines in Fig. 3) is employed as a case study due to its provision of detailed output, which was not universally available across all models. NorESM2-MM is representative for the ensemble behavior as it exhibits trends in surface stress and wind speed comparable to the multimodel mean (Supplementary Fig. 1). In terms of mean state, NorESM2-MM simulates surface stress and wind speeds at the lower end of the spectrum (Fig. 3a–d). This is likely attributed to its exceptionally large sea ice cover (see Section **Changing ice concentration, thickness, and speed**). NorESM2-MM exhibits an unrealistically large sea ice area

**Table 1 | Characteristics of the 16 CMIP6 models used in this study: horizontal resolution in the Arctic, atmospheric model component, ocean model component, sea ice model component, ice-ocean drag coefficient, and reference**

| Model | Resolution | Atmospheric model | Ocean model | Sea ice model | $10^{-3} C_w$ | Reference |
|---|---|---|---|---|---|---|
| CAMS-CSM1-0 | 54 km | ECHAM5-CAMS | MOM41 | SIS1.0 | 3.24 | 68 |
| CanESM5 | 50 km | CanEM5 | NEMO3.4.1 | LIM2 | 5.0 | 69 |
| CESM2 | 41 km | CAM6 | POP2 | CICE5 | 5.36 | 70 |
| CMCC-ESM2 | 49 km | CAM5.4 | NEMO3.6 | CICE4 | n/a | 71 |
| CNRM-CM6-1 | 49 km | ARPEGE | NEMO3.6 | GELATO | 10 | 72 |
| EC-Earth3 | 49 km | IFS 36r4 | NEMO3.6 | LIM3 | 5.0 | 73 |
| FIO-ESM-2-0 | 50 km | CAM5 | POP2 | CICE4 | 5.36 | 74 |
| GFDL-CM4 | 9 km | AM4 | MOM6 | SIS2 | 3.24 | 75 |
| GFDL-ESM4 | 18 km | AM4 | MOM6 | SIS2 | 3.24 | 76 |
| HadGEM3-GC31-LL | 49 km | GA7 | NEMO3.6 | CICE5.1 | 10 | 77 |
| IPSL-CM6A-LR | 49 km | LMDZ6A | NEMO3.2 | LIM3.6 | 5.0 | 78 |
| MIROC6 | 39 km | AGM | COCO4.9 | COCO4.9 | 20 | 79 |
| MPI-ESM1-2-HR | 36 km | ECHAM6.3 | MPIOM1.63 | MPIOM1.63 | 4.5 | 80 |
| MRI-ESM2-0 | 39 km | MRIAGCM3.5 | MRI.COMv4 | MRI.COMv4 | 5.5 | 81 |
| NorESM2-MM | 38 km | CAM6-Nor | BLOM (MICOM) | CICE5 | 5.36 | 29 |
| UKESM1-0-LL | 50 km | MetUM-GA7 | NEMO3.6 | CICE5 | n/a | 82 |

The horizontal resolution in the Arctic (2nd column) was calculated as the square root of the total area north of 70°N divided by the number of points the model has north of 70°N. Parameters used in this study include: *tauuo, tauvo, siconc, sithick, sivol, siv, siu, sfcwind, tauu, tauv, siforceinstrx, siforceinstry, sistrxdtop, sistrxubot, sistrydtop, sistryubot, tas, psl, uo, and vo*. All models employ a constant ice-ocean drag coefficient $C_w$, with values given above.

throughout all seasons, a characteristic also noted in previous studies[29]. However, we contend that the general behavior and mechanistic understanding gained from NorESM2-MM remain applicable across the models, and as demonstrated in the subsequent sections, an overestimation of sea ice area likely results in an underestimation of our main results (increase in surface stress and its subsequent impact on the ocean).

## Changing wind vs. changing ice

In regions with partial ice cover, the total ocean surface stress is commonly computed as a concentration-weighted average[13,35], considering the separate fluxes at the ice-ocean boundary and the atmosphere-ocean boundary: $\vec{\tau}_o = (1-A)\vec{\tau}_{ao} + A\vec{\tau}_{io}$, where $A$ is the sea ice concentration (percentage of ice area relative to the total area of grid cell), and $\vec{\tau}_{ao}$ and $\vec{\tau}_{io}$ represent the atmosphere-ocean and ice-ocean stress, respectively. The sea ice model is forced by the atmosphere-ice stress, $\vec{\tau}_{ai}$, and the ice-ocean stress, $\vec{\tau}_{io}$. The ocean model is forced by $\vec{\tau}_o$ (which is computed from $\vec{\tau}_{io}$ and the atmosphere-ocean stress, $\vec{\tau}_{ao}$). These are all typically computed using a bulk formula ref. 15, and the references therein. Although more advanced formulations exist (as will be elaborated on in Section **Reduced dampening effect**), all the CMIP6-deck simulations in this study parameterized the ice-ocean momentum transfer based on a constant drag coefficient, with values spanning a wide range from $3.24 \times 10^{-3}$ to $20 \times 10^{-3}$ (Table 1).

We employ NorESM2-MM to decompose ocean surface stress trends into contributions from total atmosphere-ocean stress, $\int_{\text{Arctic}} (1-A)\vec{\tau}_{ao}\, dS$, and total ice-ocean stress $\int_{\text{Arctic}} A\vec{\tau}_{io}\, dS$. 'Total' in this context refers to the area integral, thus the combined effect of changing sea ice area $A$ and changing stress. The monthly trends in total ocean surface stress (thick grey bars, Fig. 4a) exhibit the largest changes in winter (November-March) and the smallest in summer (June-August), consistent with Fig. 2a.

The positive trends in total ocean stress are primarily attributed to increased total atmosphere-ocean stress, which exhibits an increase throughout all months (teal bars, Fig. 4a). This is caused by a combination of reduced ice area (decreased $A$, light green bars Fig. 4b) and increased atmosphere-ocean stress ($\vec{\tau}_{ao}$, light blue bars Fig. 4b) due to increasing wind speed. Reduced ice area enhances momentum

transfer as, in NorESM2-MM (and all other models), open water stress (atmosphere-ocean) is larger than ice-ocean stress (detailed in Section **Reduced dampening effect**). The effect of increasing open-water area (diminishing sea ice area) dominates during early summer (May-August), whereas changing wind has the larger influence in fall and early winter (September-February).

The total ice-ocean momentum flux (magenta bars, Fig. 4a) decreases during the summer, fall, and early winter months (June-January), resulting in an overall negative contribution to the trend in total ocean stress. Primarily, this decrease is caused by a decrease in sea ice area $A$. However, during late winter and spring (February-May), the total ice-ocean stress increases despite a reduction in sea ice area. This increase in total ice-ocean stress amplifies the positive trend in atmosphere-ocean stress, in contrast to the dampening observed during the rest of the year (Fig. 4a). In February the increased ice-ocean stress explains approximately 29% of the trend, whereas in March and April it explains approximately 50%. In the following, we explain how the increased total ice-ocean stress during winter can be explained through reduced dissipation of atmospheric stress in the ice pack.

The total ice-ocean stress can be decomposed into $\int_{\text{Arctic}} A\vec{\tau}_{io}\, dS = \int_{\text{Arctic}} (A\vec{\tau}_{ai} - AF_i)\, dS$, where $F_i$ denotes the ice internal stress[13] and $\vec{\tau}_{ai}$ the atmosphere-ice stress. The latter is the divergence of the ice internal stress tensor given by the sea-ice rheology, which describes the relationship of resisting ice strength, ice motion, and large-scale deformation[36]. Although detailed formulations of sea-ice rheology differ across the models, overall, the internal stress acts like a sink of momentum that is transferred into the ice pack. As sea ice concentration decreases in a warming climate, the total internal stress in the ice pack $AF_i$ is reduced (Fig. 5c), and less energy is dissipated to ice deformation. Since internal stress is subtracted from the total atmosphere-ice stress (see equation, Fig. 4d), a reduction of total internal stress (Fig. 4c) means a positive contribution to the trend in total ice-ocean stress (pink bars, Fig. 4d). Effectively, a negative trend in internal stress is a positive trend for ice-ocean stress, as it implies a more direct transfer of atmospheric momentum to the ocean. The trend in total atmosphere-ice stress is negative in all months, primarily due to the reduction of ice area (yellow bars, Fig. 4d). However, in winter (Feb to May), the reduced dissipation of momentum into internal stress dominates the response, such that the trend in total ice-

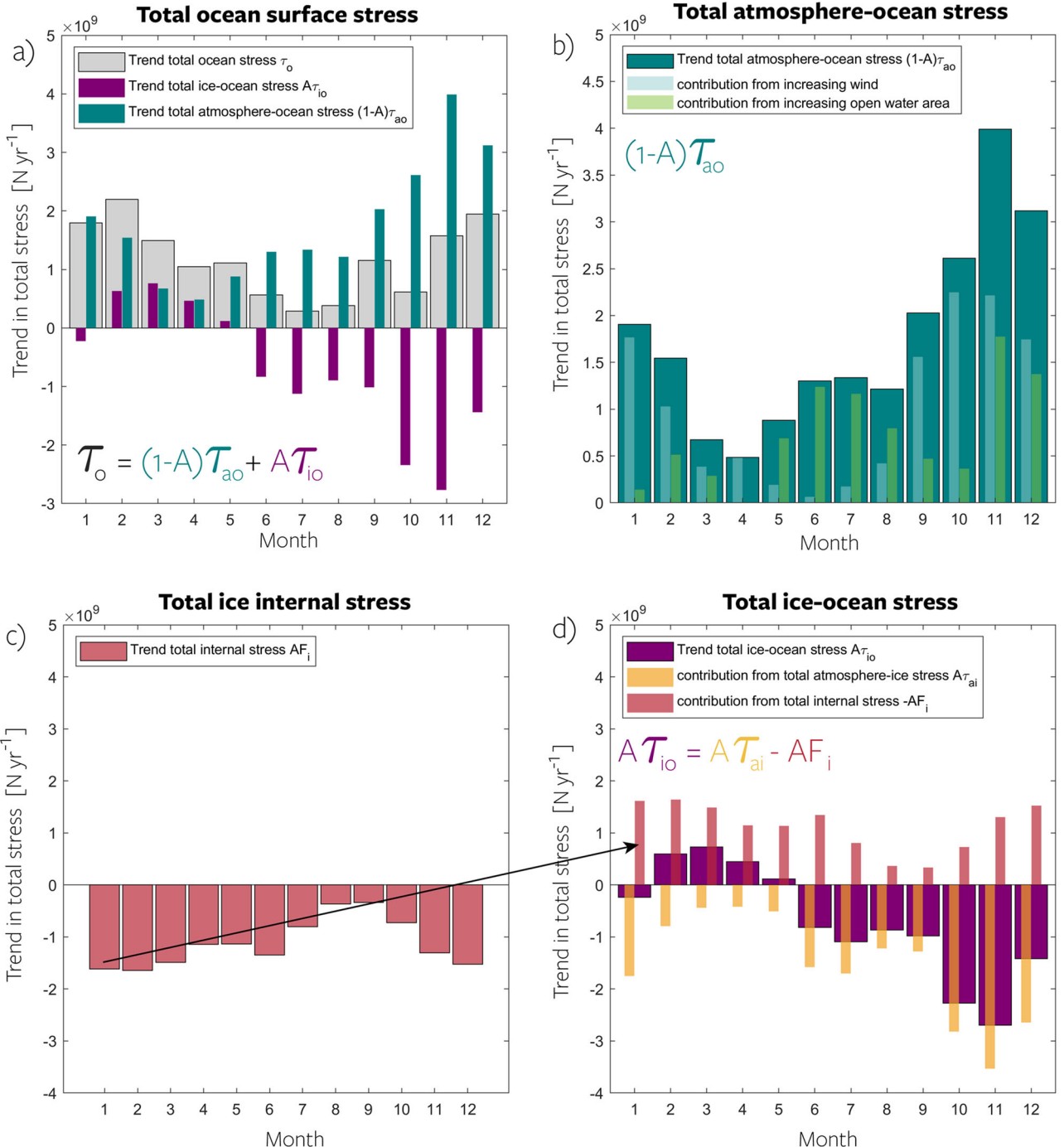

**Fig. 4 | Decomposed contributions to changes in ocean stress. a** Monthly trends from 2015 to 2100 in total area-integrated ocean surface stress in the NorESM2-MM model (Methods), showcasing the relative contributions of total ice-ocean stress (magenta bars) and atmosphere-ocean stress (teal bars). **b** Same as for **a** but for the total atmosphere-ocean stress decomposed into increasing wind contribution (light blue bars) and increasing open water area contribution (light green bars).

**c** Same as for **a** but for the total internal stress. **d** Same as for **a** but for the total ice-ocean stress (magenta bars in **a** and **b**) decomposed into contributions from total atmosphere-ice stress (yellow bars) and contributions from internal stress (pink bars). The sign of the pink bars from **c** is flipped in **d** because a negative trend on a negative term results in a positive contribution to the ice-ocean stress. Source data are provided as a Source Data file.

ocean stress becomes positive, despite the reduced momentum uptake of the sea ice from the atmosphere.

This mechanism, whereby a weakening of the ice pack results in reduced internal stress, leading to amplified ocean surface stress, is analogous to that demonstrated by Martin et al.[13] in their historical simulations. However, it is much stronger in our simulations due to the large reduction in sea ice driven by future global warming. This is further illustrated by the time series of the different stress terms for a

summer month (July) and a winter month (February), shown in Supplementary Fig. 2. Considering the similarity in ocean surface stress trends and understanding the principle physics implemented, we believe it is likely that the described mechanism is the dominant cause for the robust winter ocean surface stress amplification and seasonal variations at play across all models (Fig. 2a).

In summary, the total ocean surface stress experiences an increase across all months. This increase results primarily from a combination

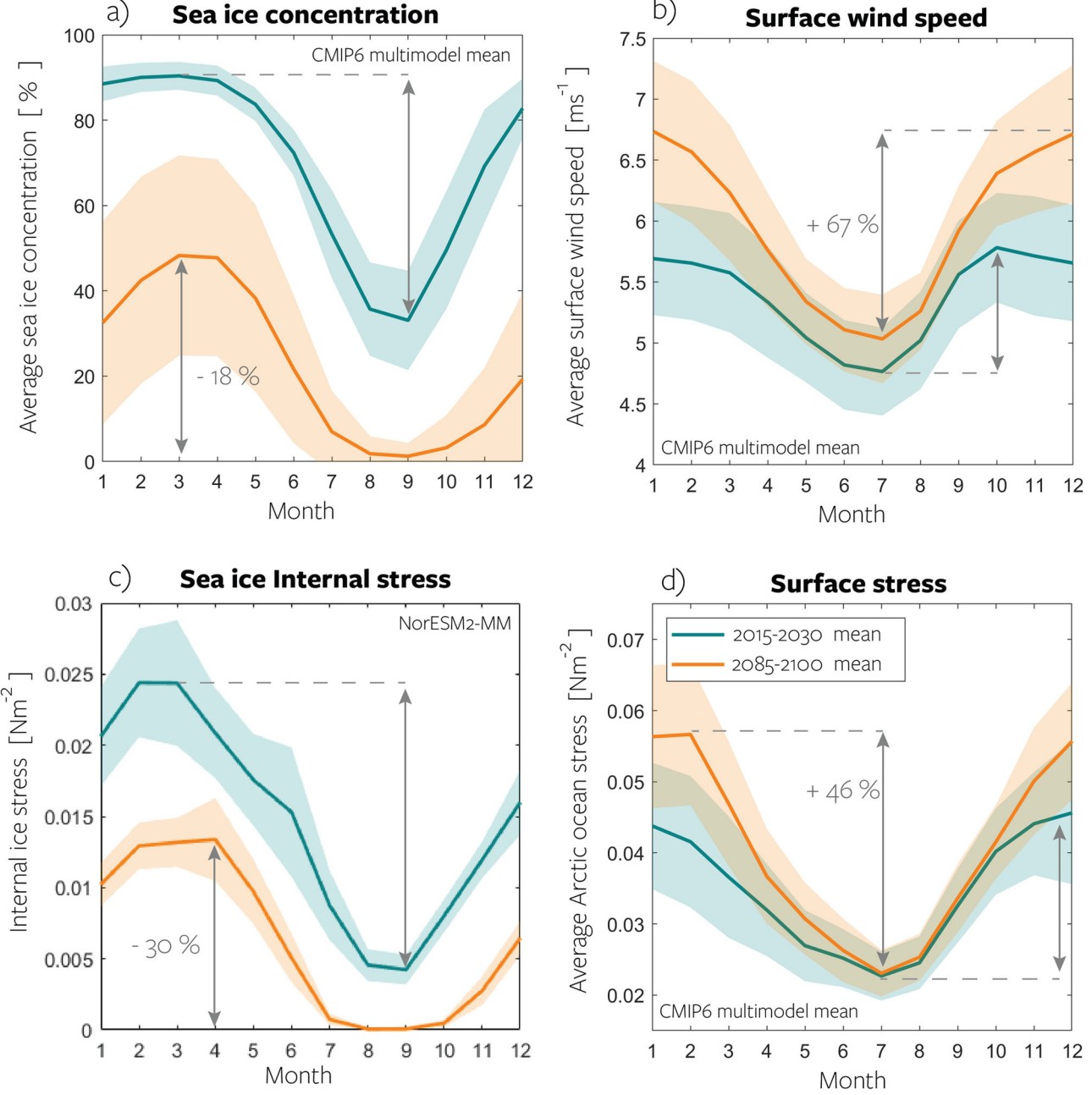

**Fig. 5 | Seasonal change at the Arctic Ocean's surface.** Monthly averages of Arctic Ocean properties in the early 21st century (teal lines), mid-21st century (dashed orange lines), and late 21st century (solid orange lines) based on the CMIP6 multimodel mean. Panels include **a** sea ice concentration, **b** surface wind speed, **c** sea ice internal stress, and **d** surface stress. Shading in panels **a**–**d** represents the standard deviation of the CMIP6 model spread, while panel **c** exclusively shows NorESM2-MM and the shading represents the interannual spread for each month. Arrows indicate the timing and amplitude of the seasonal maximum and minimum, and percentages depict the change in seasonal amplitude from 2015–2030 to 2085–2100. Source data are provided as a Source Data file.

of declining sea ice area (ice dampens, while open water presents larger drag, Section Reduced dampening effect) and increasing wind speed. Averaged across all months, both factors contribute approximately evenly (Fig. 4). The total ice-ocean stress decreases in most months, but during winter, it increases because less momentum is absorbed by ice internal stress resulting in a more direct transfer from the top to the bottom of the ice.

### Changing seasonal disparities
Martin et al.[13] observed a pronounced change in the seasonality of ocean surface stress from 1979 to 2012, characterized by an increase during fall and winter and a decrease during summer. However, the atmospheric forcing of their simulations showed no trend in wind speed for the given period. Our results reveal large projected changes in the seasonality for wind speed, stress, and sea ice concentration in the first decades of the projections (Fig. 5).

Based on the CMIP6 multimodel ensemble projections, we find both for the early and late 21st century, the annual cycle of sea ice concentration has a minimum (37% and 2%, respectively) in September and a maximum (91% and 48%) in March (teal and orange lines, Fig. 5a). The amplitude of the annual cycle is reduced by 18% from 2015 to 2100.

For wind speed, the amplitude of the annual cycle increases (67%, Fig. 5b). Winds are generally weakest in early summer and strongest in

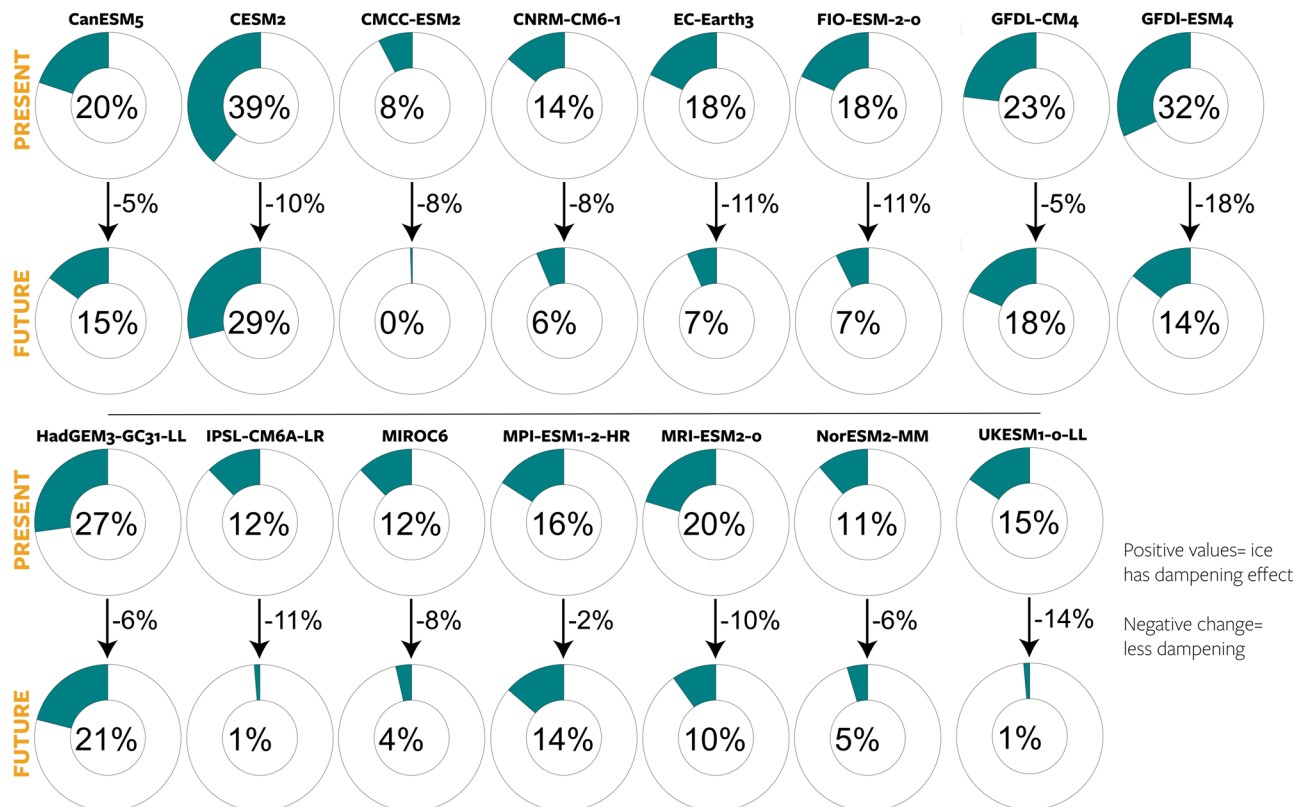

**Fig. 6 | Integrated effect of sea ice on momentum transfer.** Doughnut charts indicate the total dampening effect (energy loss) of ice on the transfer of atmosphere to ocean stress (Methods) during the current period (2015–2030) and projected future (2085–2100). Arrows show the climate change response, where negative values indicate a reduced dampening effect. CAMS-CSM1-0 was excluded from this analysis due to unavailable atmospheric data. Source data are provided as a Source Data file.

fall and winter. The substantial trend during December and January, in contrast to minimal change during summer, intensifies this seasonal disparity. This is in agreement with DuVivier et al.[30]. Moreover, the peak shifts from October to December, leading to a 3-month phase shift in the annual cycle of the multimodel ensemble mean surface winds.

Internal ice stress, only available for NorESM2-MM (Fig. 5d), exhibits a seasonal cycle mirroring sea ice concentration, with the largest change expected during winter since it cannot decrease further during summer. Here, the overall seasonal cycle diminishes (−30%) as the ice concentration decreases.

Towards the late 21st century, model ensemble mean ocean stress (Fig. 5d) increases in all months except July, increasing the seasonal amplitude by 46%. In contrast to wind speed, the increase is largest during February (related to the large change in internal stress that month), shifting the peak stress from December to February. Thus, in line with previous studies[13], the models predict a significant alteration in the seasonality of Arctic surface stress.

**Reduced dampening effect**

The momentum transfer from the atmosphere to the ocean and sea ice, as well as from sea ice to the ocean, varies across models due to differences in the bulk formulae, specifically in drag coefficients at air-ice and ice-water interfaces.

To assess the variability associated with the wide range of drag coefficients (Table 1) and momentum transfer formulations in the models, we introduce the "dampening effect," inspired by the "amplification index" employed by Martin et al.[15]. This metric, calculated as $100\% \times (\int_{\text{Arctic}} \vec{\tau}_a \, dS - \int_{\text{Arctic}} \vec{\tau}_o \, dS)/\int_{\text{Arctic}} \vec{\tau}_a \, dS$, estimates energy loss by comparing the total integrated atmospheric wind stress

to the total integrated ocean surface stress (depicted as doughnut charts in Fig. 6). Positive values indicate that the sea ice has a general dampening effect (atmospheric wind stress exceeds the ocean surface stress); for example, in CanESM (first model, Fig. 6), the total ocean surface stress is on average 20% less than the total atmospheric stress.

Despite substantial inter-model differences, all models indicate an overall dampening effect ranging between 8% and 39% (Fig. 6). Notably, CESM2 and GFDL-ESM4 exhibit particularly strong dampening effects, while CMCC-ESM2 and MIROC6 show very weak dampening effects. Interestingly, there appears to be no direct relation between the average dampening effect and the drag coefficient (Table 1), suggesting that the mean ice state likely plays a crucial role as well. However, the dampening effect is also evident in Fig. 3, where we observe that CESM2 and GFDL-ESM4 exhibit average wind speeds but particularly low ocean surface stress. In contrast, CMCC-ESM2 and MIROC6 demonstrate particularly high ocean surface stress.

All models exhibit a consistent climate response (negative), indicating a reduced dampening effect of sea ice in the future. Furthermore, the magnitude of this reduction falls within a relatively narrow range of −2% to −18%, underscoring a robust response. The multimodel mean depicts an initial dampening effect of 19% at the beginning of the century, which decreases by 9% towards the end of the century.

While additional diagnostics would be needed to assess whether the partition of the various stress components is consistent across the different models, another large uncertainty lies in the drag formulation itself. Unfortunately, more advanced parameterizations, e.g. of the ice-state-dependent form drag, that are available in some models[14], are still subject to substantial uncertainty[7,37–39], with previous studies demonstrating that even the sign (increasing or

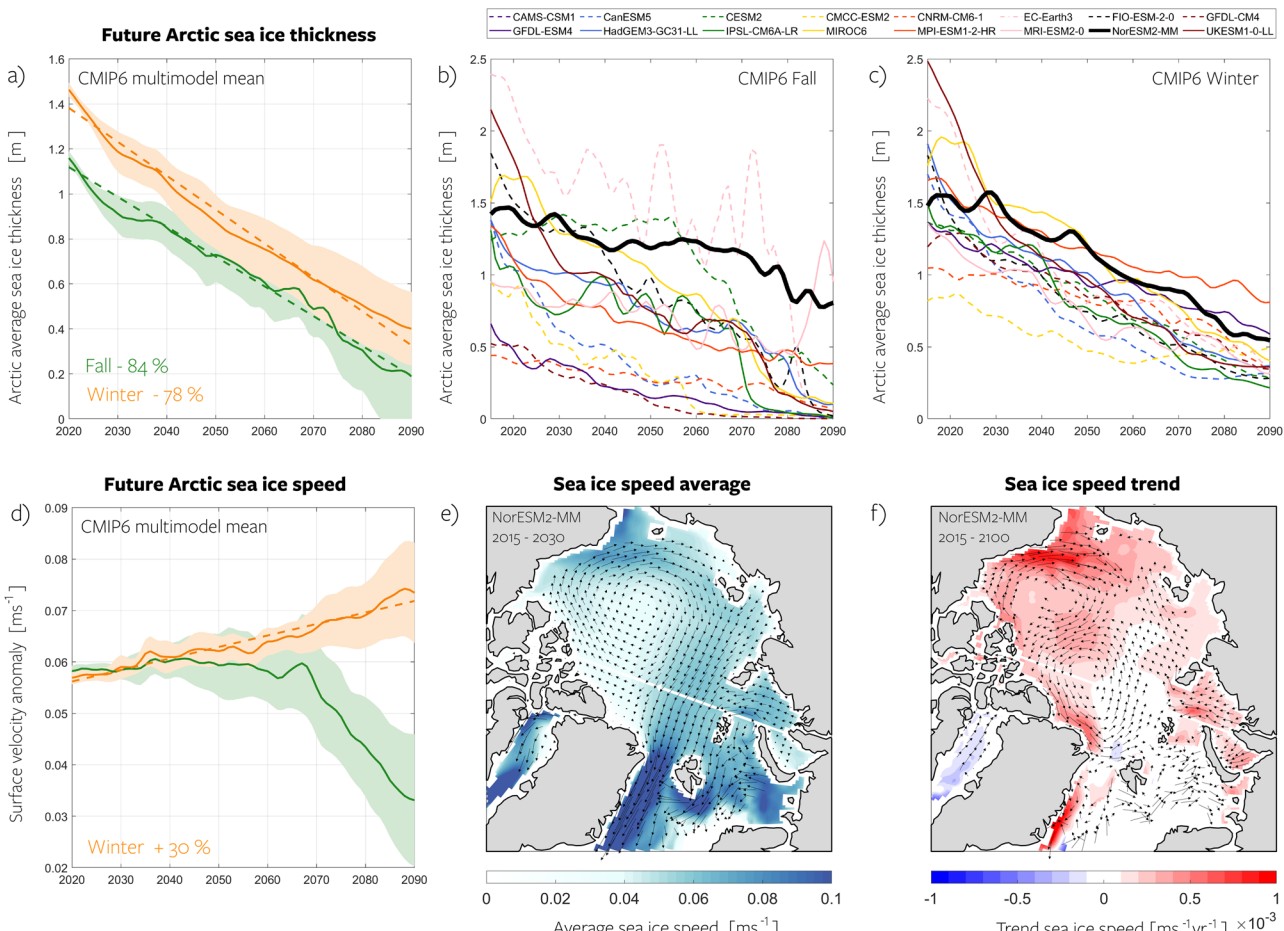

**Fig. 7 | Changing ice thickness and speed. a** multimodel mean Arctic Ocean sea ice thickness anomalies (relative to 2015–2030 model mean) projected by the CMIP6 models under a high-emission scenario for fall (green) and winter (orange). Envelopes indicate the model spread as determined by one standard deviation, and values indicated in the panel represent the seasonal increase from 2015 to 2100. Time series represent a basin-wide average and have been smoothed using a low-pass filter with a five-year cutoff frequency. The dashed lines represent the linear trend for winter and fall until 2100. **b** Spaghetti plot of fall sea ice thickness time series for all CMIP6 models. **c** Same as **b** but for winter. **d** Same as **a** but for sea ice velocity. **e** Spatial representation of the annual mean sea ice speed from NorESM2-MM, serving as an example. **f** Linear trend in annual mean sea ice speed from NorESM2-MM. CESM2 and CMCC-ESM2 were excluded from the sea ice speed analysis due to unavailable data. Source data are provided as a Source Data file.

decreasing) of calculated stress is sensitive to the details of stress parameterization[15,37].

Although more advanced drag formulations are expected to alter the total stress response, we have shown that the primary driver of the trends is the increasing wind speed, which impacts all stress components. Additionally, a future thinning of the ice pack will result in a weakening of the ice. Thus, despite uncertainties in parameterizations, it is reasonable to conclude that overall Arctic Ocean surface stress will increase due to the combination of increasing winds and reduced internal stress, and the estimates provided by the CMIP6 models represent the best available at this point.

**Changing ice concentration, thickness, and speed**
In addition to differences in parameterization, variations in sea ice states and rates of change influence both the mean and trend in surface stress. For instance, quantitatively, models with a small total sea ice area exhibit relatively high mean surface stress (see Fig. 3). Martin et al.[13] explored the relationship between surface stress and sea ice concentration in detail, finding a stress peak at approximately 80% ice concentration. We conducted a similar analysis across all CMIP6 models (Supplementary Fig. 4). Each model's sea ice concentration (for grid cells with more than 0% ice) was binned with a width of 2%. Within

each bin, ocean surface stress was plotted against sea ice concentration. Additionally, the results were normalized by wind speed to remove the effects of spatial and temporal variations in winds. Our findings are not as definitive as those reported by Martin et al.[13] and need to be treated with caution. Their analysis was conducted using daily data, whereas our model data is available only as monthly averages. Moreover, over 80% of grid cells (in all models) have sea ice concentrations either above 95% or below 10%, which affects the robustness of the relationship for intermediate ice concentrations. This, combined with the coarse spatial resolution of the models, provides a limited statistical foundation for this analysis. Taking this into consideration, our results do not exhibit a clear stress peak at 80%, but they do suggest a maximum ocean surface stress for ice concentrations between 10 and 90%. Since most models show a gradual increase in stress with rising sea ice concentration, followed by a decrease towards 100% ice concentration (consistent with Martin et al.[13]), it can be hypothesized that if more grid cells had lower ice concentration, the average stress would be lower. We note that if the grid cell is completely ice-free, stress can be higher again.

In addition to ice concentration, sea ice thickness plays a pivotal role as it determines the strength of the ice. Sea ice thickness shows a decrease of 85% during fall and 78% during winter towards the end of

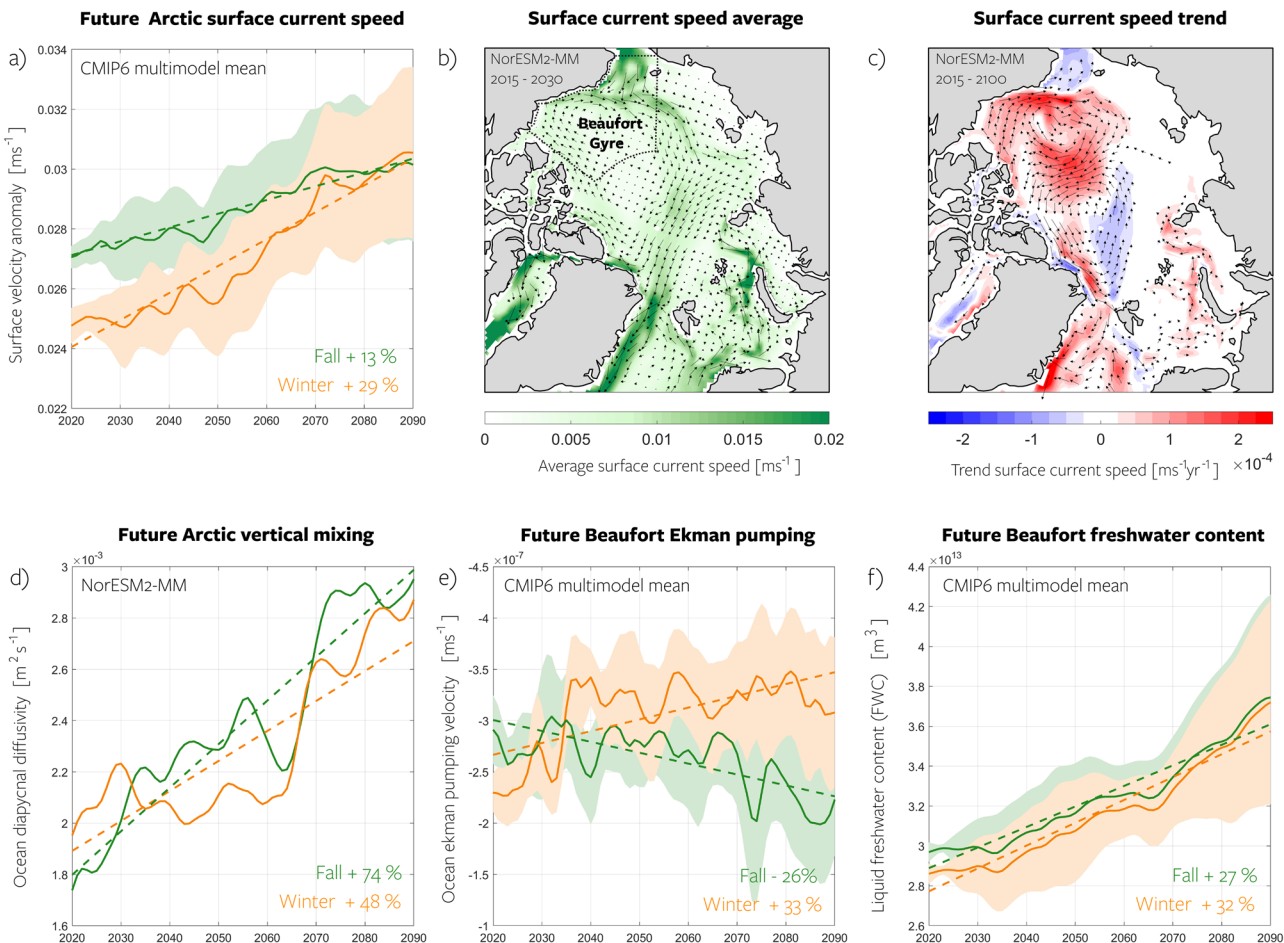

**Fig. 8 | Impact on ocean dynamics. a** Multimodel mean Arctic Ocean surface current speed (Methods) anomalies (relative to 2015–2030 model mean) projected by the CMIP6 models under a high-emission scenario for fall (green) and winter (orange). Envelopes indicate the model spread as determined by one standard deviation, and values indicated in the panel represent the seasonal increase from 2015 to 2100. Time series represent a basin-wide average and have been smoothed using a low-pass filter with a five-year cutoff frequency. Dashed lines represent the linear trend until 2100 and individual model trends are shown in Supplementary

Fig. 1. **b** Spatial representation of the annual mean ocean surface velocity NorESM2-MM, serving as an example. **c** Linear trend in annual mean ocean surface velocity for NorESM2-MM. **d** Same as **a** but for NorESM2-MM diapycnal diffusivity averaged down to 100 m (Methods), a variable representing vertical mixing. **e** Same as **a** but for Ekman pumping (Methods) in the Beaufort Gyre region (marked by black dashed lines in **b**. **f** Same as **a**, but for liquid freshwater content in the Beaufort Gyre region. GFDL-ESM4 was excluded from the ocean surface current speed analysis due to unavailable data. Source data are provided as a Source Data file.

the century (Fig. 7a), which supports the findings in Section **Changing wind vs. changing ice**. While inter-model differences exist in average thickness, ranging between 0.9 m and 2.0 m during winter and 0.4 m and 2.0 m during fall, the rate of decline is consistent among models (Fig. 7b, c). The reduction in thickness, combined with the decrease in concentration, contributes to the decline in internal stress (Section **Changing wind vs. changing ice**). Unfortunately, since internal stress terms are unavailable, isolating this effect is not feasible, and separating the effects of changing thickness and ice concentration on internal stress is not possible, but the observed trends support the overall picture provided by the NorESM2-MM case study. If form drag were included, the effect of sea ice thickness changes would be even more significant, as it would also directly affect the drag coefficients.

In addition to influencing stresses, the combination of changing ice concentration, thickness, and wind speeds also impacts the drift of sea ice. During winter and in the annual mean, the average drift speed increases throughout the Arctic, consistent with stronger winds and a weaker sea ice cover (see Fig. 7d–f). The CMIP6 multi-model mean projects a 30% increase in sea ice speed towards the end of the century. As with the other effects, the increase in ice speed could be a response to changing winds or changing ice strength.

Previous studies have used the ratio of ice speed to wind speed as a proxy for ice rheology[19,40], but such analysis is beyond the scope of this study. We here calculate sea ice speed only where sea ice is present, and the decline in sea ice area impacts these results. The multimodel mean sea ice speed during fall (see Fig. 7d) initially increases until around 2040, then gradually decreases. This is due to the rapid decline in sea ice area during fall, leading to averaging over fewer grid cells and potentially excluding areas with higher wind and ice speeds near the marginal ice zone.

## Impact on ocean dynamics
Increased ocean surface stress in a warming climate strengthens upper ocean circulation and results in a more energetic Arctic Ocean (Fig. 8). A statistically significant increase in ocean surface velocity is projected across the Arctic, as indicated by significant trends in nearly all models Supplementary Fig. 1). The winter amplification driven by a weakening of the ice pack is also evident in the ocean's response, with a multi-model mean trend of $8.4 \times 10^{-4}$ ms$^{-1}$ per decade during winter and $3.3 \times 10^{-4}$ ms$^{-1}$ per decade during fall (Fig. 8a). This results in a cumulative increase in surface velocity of 29% and 13% by 2100 for winter and fall, respectively.

The ocean's response is not limited to the surface; mixing in the ocean efficiently redistributes momentum downward. In climate models, mixing is typically parameterized[41]. Most CMIP6 models do not archive it, but in NorESM2-MM, we can access total ocean diapycnal diffusivity, representing the sum of all mixing components (wind, shear, tidal, etc.). In this model, diapycnal diffusivity indicates a major increase in vertical mixing by 74% in fall and 48% in winter by 2100 (Fig. 8d). While ocean currents accelerate more in winter, vertical mixing intensification is greater in fall than in winter, likely because stratification is weakest with massive surface heat loss and sea ice formation occurring in fall. This will have a profound impact on biology, as mixing influences the vertical distribution of nutrients[42] – the key limiting factor for primary production which serves as the foundation of the ecosystem[43]. Additionally, enhanced turbulence can intensify upward mixing of heat from intermediate depths[44,45], promoting further ice melt or delay refreezing[10,46] and thereby establishing a positive feedback loop that enhances ice loss in a warming climate. However, this mechanism is opposed by a future strengthening of the thermohaline stratification[47], which limits vertical mixing. For the other CMIP6 models, total kinetic energy (Methods) works as a proxy for the effect that the increased momentum input may have on vertical mixing, and exhibits a consistent increase across the model suite (Supplementary Fig. 3).

An additional crucial impact is the influence on the dynamics of the Beaufort Gyre. The Beaufort Gyre has been demonstrated as a region where the ocean response to wind forcing is highly mediated by changes in sea ice internal stress through a mechanism dubbed the ice-ocean governor[18,48,49]. Indeed, the spatial patterns of surface currents and their changing speed as illustrated by NorESM2-MM (Fig. 8b, c), highlight that the strongest positive trends are simulated in the Beaufort Gyre region. The Beaufort Gyre contains a large reservoir of liquid freshwater[50,51], and its dynamics are key to the climate system[52,53]. Beaufort Gyre freshwater content is regulated by surface wind stress, which drives the Ekman convergence and halocline deepening[54], and previous model experiments have demonstrated that increased stress leads to freshwater content (FWC) increase[55], although different sea ice conditions mediate this response[18,56]. Consistently, we find that in future CMIP6 simulations, enhanced surface stress affects Beaufort Gyre Ekman pumping and FWC (Fig. 8e, f). FWC increases in all seasons in all models. However, we note that this change is not solely attributable to variations in stress; it is primarily driven by increased freshwater fluxes[53,57], including runoff, precipitation, ice melt, and Bering and Barents Sea inflows. Surface stress increase enhances this trend by containing it within the Gyre. The effect of enhanced Ekman pumping is limited to the winter season (+33%), with the multimodel mean showing a negative trend in Ekman pumping during fall (-26%). This observed seasonal variation could be attributed to the fact that winter exhibits the highest trend in surface stress. Given that this intensified winter trend in surface stress is related to reduced energy loss to internal stress (Section **Changing wind vs. changing ice**), the Ekman pumping is indirectly affected by internal stress reduction. Our findings support the hypothesis of a potentially less effective ice-ocean governor in the future. However, we do not definitively prove this assertion, as it is beyond the scope of this paper to delve into the intricacies of the ice-ocean governor and its changes in the individual models.

## Discussion

State-of-the-art climate models consistently predict an increase in Arctic Ocean surface stress due to a combination of increased surface wind speeds, reduced sea ice concentration, and a weaker ice pack, as illustrated in Fig. 1b. Increased wind speed is the primary driver of trends in ocean surface stress. However, while the largest increase in wind speeds is projected for the fall, the most pronounced trend in surface stress occurs during the winter. This discrepancy is due to the

modulation of momentum transfer by the changing sea ice state. We find that sea ice consistently dampens ocean surface stress in all CMIP6 models and that this effect is reduced as sea ice declines in a projected warmer climate. In our NorESM2-MM case study, the dampening is reduced the most in winter, because of the diminishing role of dissipation of momentum by ice deformation that allows more of the wind's momentum to be transferred into the ocean. This effect explains 29−50% of the model's winter trend in surface stress, highlighting the important role of reduced ice internal stress in a declining ice pack, and exemplifying how the total surface stress responds to an interplay of multiple components. Figure 1b illustrates that the enhanced surface stress results in a more energetic Arctic Ocean with strengthened surface velocity and enhanced vertical mixing. Vertical mixing plays a pivotal role in the Arctic, as it directly influences phytoplankton growth through its impact on the vertical nutrient flux[42]. With enhanced mixing, we can thus expect a continued rise in primary production, a trend already observed in recent decades[58]. Enhanced stress further strengthens Ekman transport, exerting a substantial influence on the Beaufort Gyre convergence, ultimately contributing to a growing liquid freshwater content. These changes in Beaufort Gyre dynamics affect freshwater pathways[50], and hence ocean circulation in the North Atlantic[59], posing severe potential implications for global climate[3] and Arctic communities[60].

While the overall response of enhanced surface stress remains robust across climate models, the specifics of how sea ice moderates the momentum flux depend on model formulations and the (simplified) formulation of ice-ocean drag, introducing considerable uncertainties. A shift to a less deformed and more mobile ice pack in the Arctic Ocean has already been observed[5] and there is an urgent need to enhance the coupling of these processes in numerical models and to perform more simulations with a realistic representation of the sea ice internal stress and more advanced form drag formulations[15,37]. Due to the lack of pan-Arctic observations, parameters involved are poorly constrained[14,37], but we know there is significant spatial and temporal variability in sea ice surface roughness[61–64], and bottom roughness[6,7,37], which should be incorporated into models. These parameters are, however, expected to evolve in response to climate changes, adding complexity.

In conclusion, our results demonstrate the profound impacts of projected changes in the Arctic Ocean momentum budget, highlighting the importance of additional observations at the atmosphere-ice-ocean interface for a comprehensive understanding of the future evolution of the Arctic Ocean.

## Methods
### Climate model data
We use monthly mean model output obtained from 16 models (Table 1) participating in the Climate Model Intercomparison Project phase 6 (CMIP6)[28]. The models used in this study are consistent with those employed and evaluated for the Arctic region by Heuzé et al.[65], with the inclusion of CMCC-ESM2, CNRM-CM6-1, and HadGEM3-GC31-LL. The models were selected from the CMIP6 models used in Heuzé et al.[66] to represent a diverse range of vertical grid types and were chosen after excluding models with poor bathymetry[65]. Typical horizontal model resolution is ~ 50 km in the Arctic (9 km for the highest resolution). The output we use are the surface wind speed 'sfcwind', sea ice concentration 'siconc', sea ice thickness 'sithick' or 'sivol', eastward and northward ocean surface stress 'tauuo' and 'tauvo', eastward and northward atmospheric wind stress 'tauu' and 'tauv', eastward and northward ice velocity 'siu' and 'siv' and eastward and northward ocean velocity 'uo' and 'vo'. Only 5 models provide the stresses at the top ('sistrxdtop' and 'sistrydtop') and the bottom ('sistrxubot' and 'sistryubot') of the ice enabling us to differentiate between ice-ocean and atmosphere-ice stress. Internal stress in the ice pack ('siforceintstrx' and 'siforceintstry') was only archived for

NorESM2-MM. The latter is therefore used to derive a full momentum energy budget (Fig. 4). Vertical diapycnal diffusivity ('difdia') was also only archived for NorESM2-MM. All computations were performed on the models' native grid. We evaluated the first 85 years of the future high (ssp585) emission scenario[28], i.e., January 2015–December 2100. We specifically chose the high-emission scenario to effectively distinguish climate change signals from internal variability. No other emission scenarios are included because the focus of this study is the mechanistic understanding of a changing Arctic, not the differences between different scenarios. Furthermore, for winter (which is the primary focus season of our study), more than 85% of the projection uncertainty is due to model differences, not the emission scenario (see Fig. 1d in Bonan et al.[31]). For each model, only one ensemble member was used: 'r1i1p1f1' for the majority of models; 'r1i1p1f2' when r1i1p1f1 was not available (CNRM-CM6-1 and UKESM1-0-LL), and 'r1i1p1f3' for HadGEM3-GC31-LL. We note that simulated internal variability is not investigated in detail but argue that under the high-forcing scenario, this is significantly less than the forcing-driven trends (not shown).

## Statistical metrics

In this study, all metrics are computed as weighted averages per grid cell area or integrated totals across the entirety of the Arctic Ocean. For our purposes, the Arctic Ocean encompasses the deep central Arctic Ocean and the shelf seas, namely the Beaufort, Chukchi, East Siberian, Laptev, Kara, and Barents seas. The boundaries of our defined Arctic Ocean region are delineated by the Fram Strait, Barents Sea Opening, Narres Strait, and Bering Strait (dashed red lines in Fig. 2c). We have computed seasonal averages defined as winter (January-March), summer (June-August), and fall (September-November). These seasons follow the annual cycle of the Arctic sea ice cover in terms of its areal coverage represented by the sea ice concentration averaged over the entire Arctic Ocean. Trends in ocean stress and surface wind speed were calculated from 2015-2060, and not over the full future period because the changes we observe are transient, and there is some flattening towards the end of the century (Fig. 2a, b). All trends presented are statistically significant at the 95% confidence level unless otherwise stated, and uncertainty was measured by the standard error computed from the monthly time series. Spatial averages and linear trends (Fig. 2c, d and Fig. 4b, c, e, f) are calculated based on monthly time series at every grid cell for NorESM2-MM only.

## Derived parameters

- Decomposed ocean surface stress in Fig. 4a has been calculated following Martin et al.[13]. For this case study, we employ the NorESM2-MM model since we have access to all stress terms from these simulations. The separate terms were not available for the entire model ensemble. In each grid cell the ocean surface stress is calculated as $\vec{\tau}_o = (1 - A)\vec{\tau}_{ao} + A\vec{\tau}_{io}$, where $A$ is sea ice concentration, $\vec{\tau}_o$, $\vec{\tau}_{ao}$ and $\vec{\tau}_{io}$ are the stresses at the ocean, atmosphere-ocean and ice-ocean interface, respectively, provided by the model. All components are multiplied with the grid cell area and integrated over the Arctic region to create a time series of the total Arctic Ocean momentum transfer. Similarly, the ice-ocean stress ('sistrxubot' and 'sistryubot') is decomposed into atmosphere-ice stress ('sistrxdtop' and 'sistrydtop') and internal stress ('siforceintstrx' and 'siforceintstry'). The decomposition of atmosphere-ocean stress is not provided by the model, but has been decomposed into increasing open water contribution and increasing wind contribution by assuming that the increasing open water area must explain what is not explained by the increasing wind. The wind contribution is calculated by employing the same bulk formulation as is used by the model over open water. NorESM2-MM uses a bulk formulation following Large and Pond[67]:
$C_D = (2.7|\vec{\mathbf{u}}_a|^{-1} + 0.142 + 0.0764|\vec{\mathbf{u}}_a|)10^{-3}$   and   $\vec{\tau}_{ao} = C_D|\vec{\mathbf{u}}_a|\vec{\mathbf{u}}_a\rho_a$,

where $C_D$ is the neutral drag coefficient over open water, $u_a$ the surface wind speed, and $\rho_a$ the air density at the surface. This relationship accounts for higher drag at very low wind speeds due to small-scale turbulence under calm conditions and for increasing surface roughness with increasing wind speed. We use this formulation to calculate the theoretical wind stress over the entire Arctic Ocean, as if it were open water, and calculate the linear trends based on this.

- Ekman pumping velocity is calculated from the curl of surface stress ($\vec{\tau}_o$) and is given as:

$$W_{ek} = \nabla \times \frac{\tau_o}{\rho_0 f}. \tag{1}$$

where $f$ the Coriolis parameter, and $\rho_0$ a reference density of 1027.8 kgm$^{-3}$. Negative Ekman pumping values denote downward velocities (convergence). Both GFDL models were excluded from this calculation due to problems with the calculations of the wind stress curl, likely due to wrong grid information.

- Liquid freshwater content is calculated as:

$$\text{FWC}(t) = \iint \int_{z(S=S_{\text{ref}})}^0 \frac{S_{\text{ref}} - S}{S_{\text{ref}}} dV \tag{2}$$

relative to a reference salinity ($S_{\text{ref}}$) of 34.8, following Marshall et al.[55] and Cornish et al.[56].

- Ocean kinetic energy is calculated as the integrated kinetic energy down to 100 m: $\int_{z=100}^{z=0} 0.5 \times (\mathbf{u}^2 + \mathbf{v}^2) \times m\, dz$, where $m$ is the mass of the grid cell calculated as the product of the area, layer thickness, and average seawater density (1025 kg/m$^3$).

- The dampening factor refers to the loss of wind energy caused by sea ice dynamics during the exchange between the atmosphere and the ocean. It is calculated as the difference between the integrated atmosphere wind stress and the ocean surface stress: $100\% \times (\int_{\text{Arctic}} \vec{\tau}_a\, dS - \int_{\text{Arctic}} \vec{\tau}_o\, dS)/\int_{\text{Arctic}} \vec{\tau}_a\, dS$. This methodology is comparable to the 'amplification index' by Martin et al.[15], which is a ratio of ocean surface stress divided by the (potential) wind stress on open water. When the atmospheric wind stress exceeds the ocean surface stress, the dampening factor is greater than 0%; in contrast, when the atmospheric wind stress lags behind the ocean surface stress, the factor is smaller than 0%. This means that sea ice dampens momentum transfer when the factor exceeds 0%, while it amplifies momentum transfer when the factor is smaller than 0%.

## Data availability

The datasets generated and/or analyzed during the current study are available via the Earth Grid System Federation. For the analysis presented here, we used the Geophysical Fluid Dynamics Laboratory (GFDL) node: https://esgdata.gfdl.noaa.gov/search/cmip6-gfdl/and the Lawrence Livermore National Laboratory node: https://esgf-node.llnl.gov/projects/cmip6/. Source data are provided with this paper.

## Code availability

The MATLAB code used to calculate decomposed ocean surface stress, ekman pumping velocity, liquid freshwater content, ocean kinetic energy, and the dampening factor is available at: https://doi.org/10.5281/zenodo.11190101.

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

## Acknowledgements

This study has received funding from the European Union's Horizon 2020 research and innovation programme under grant agreement no 101003826 via project CRiceS (Climate Relevant Interactions and Feedbacks: the key role of sea ice and Snow in the polar and global climate system). TH was supported by Research Council of Norway grant no. 314570. The computations were performed on resources provided by Sigma2 - the National Infrastructure for High-Performance Computing and Data Storage in Norway, grant nos. NS9081K and NN9824K.

## Author contributions

M.M. conceived the study, analyzed the data, and wrote the initial draft of the paper. T.H., T.M., and M.A.G. provided feedback on the

analysis and contributed to constructive revisions. All authors contributed to editing and revisions. M.A.G. and T.H. acquired the funding (CRiceS).

## Funding

## Competing interests
The authors declare no competing interests.
