## [Peer Review File · Nature Communications]

Future sea ice weakening amplifies wind-driven trends in surface stress and Arctic Ocean spin-upREVIEWER COMMENTS

Reviewer #1 (Remarks to the Author):

Review of “Future sea ice weakening amplifies wind-driven Arctic Ocean spin-up ” by Muilwijk et al.

In this manuscript Muilwijk and co-authors present an analysis of future changes in the Arctic. The reduction in sea ice concentration leads to an increase in the surface stress experienced by the ocean, with consequences for surface speed, freshwater storage, and diapycnal mixing. The analysis is thorough, the results are clear, and the conclusions are robust. This manuscript will make a valuable contribution to the literature. I have a few minor comments that I would like the authors to address prior to publication.

Edward Doddridge

General Comments

Summer wind speed vs stress

The summer wind speed increases linearly through time out to 2100, but the wind stress plateaus. Is there an obvious explanation for this decoupling? Is it related to the quadratic drag term in the wind stress – perhaps the wind speed is only increasing in the places with relatively slow winds, and therefore makes a negligible contribution to the total stress? I understand that this manuscript is focused on the fall and winter seasons, but this discrepancy is odd, and, in my opinion, it warrants mentioning.

Sea ice internal stress

The diagnostics shown in the manuscript highlight the role of internal ice stress. Perhaps it is a misguided notion, but I had always considered that much of the stress that the ice pack absorbs is transferred horizontally into the land boundaries surrounding the ocean. I would have appreciated a bit more discussion about what happens to the stress that is absorbed by the ice pack.

Specific comments

Line 127: “more significant” is confusing in this context. You’ve just described statistically significant trends, and I think what you mean here is ‘more substantial’ or ‘larger’. I suggest changing the wording.

Lines 153-173, Figures 2c) and 2d): There’s a lot of unused space on these panels – do we need to

see so much land? A polar stereographic projection might focus the readers' attention on the Arctic ocean and take up less space.

Line 170, Figure 2d): These figures are beautiful, but I recommend switching out the diverging colourmap in 2d) for a one sided colourmap like you've used in 2c). As far as I can tell, there is nothing particularly special about the value 0.27 N/m^2 , so highlighting areas that are higher and lower than this value isn't useful.

Line

Line 284: "Consequently, the reduction in internal sea ice stress (red bars, Fig. 3c) contributes to a positive trend in ocean stress throughout all months (gray bars, Fig. 3a)."

I suspect this is a linguistic issue when trying to describe a reduction in magnitude of a negative term, but the above sentence does not make sense to me. The red bars in Figure 3c) show a positive trend (increase) in the total internal stress, not a reduction.

Lines 388-403, Figure 5: There's a lot going on here. Well done on presenting the information so succinctly. It was not immediately obvious to me that two of the models showed a negative dampening effect – perhaps it would be helpful to plot the doughnuts, or the text labels, in a different colour for those two models. The key in the bottom right is helpful, but needs some work. `t_a` isn't described anywhere in the figure or the caption, and the 'clockwise climate response' label is plotted next to a doughnut that shows a counter-clockwise response (i.e., more dampening). (Actually, reading further in the text, I think this bit of the key is actually meant to indicate the change between two doughnuts – that wasn't clear to me when looking at the figure.)

Lines 427-8: Perhaps this result could be shown on an additional panel in Figure 5? It takes a lot of interpretation to glean it from the panels presented.

Lines 461-471: Figure 6a-c): These plot speed (a scalar), not velocity (a vector).

Lines 521-522: "This further supports the impact of reduced energy loss to internal stress." It's not clear to me how this sentence follows the previous one. Some extra guidance would be helpful.

Lines 571-574: Should this instead say "... differentiate between ice-ocean and atmosphere-ice stress"?

Lines 583-585: It is almost certainly true that internal variability is dwarfed by the forced trends, but it would add weight to your argument if you presented one time-series figure from a low emissions pathway in the supplementary material.

Line 672: I assume "love" is meant to be "over" or something similar.

Lines 721-723: The use of parentheses to denote the negative version of a sentence is convenient (difficult to interpret and frustrating) for authors (readers). I strongly suggest rewriting this sentence

so that it clearly presents each idea separately and is easier for your readers to understand.

Line 1221, Figure S2 caption: "However, during winter, the positive trend of internal stress reduction (orange line, Fig. d) outweighs the contribution of changing atmosphere-ice stress."

Two things: The total internal stress is the red line, not the orange one; and 'positive trend of internal stress reduction' is very convoluted. I recommend replacing with a simpler statement like 'the reduction in total internal stress'.

Reviewer #2 (Remarks to the Author):

This paper examines the impact of projected future changes in sea ice on surface ocean stress and the large-scale circulation of the Arctic Ocean. It draws on a suite of 16 CMIP6 runs based on a single emission scenario for part of its analysis, and utilizes the NorESM2-MM model for a more detailed analysis of the central component of the study: ice internal stresses. The study finds that diminishing ice cover will lead to increased momentum transfer from the atmosphere to the ocean, accelerating large-scale circulation and increasing turbulent mixing. While the paper also touches on potential effects on biogeochemistry, heat transport and the Atlantic Meridional Overturning Circulation (AMOC), these discussions remain speculative, and the paper does not concretely address these issues.

My major concern is that the main finding, an increase in momentum transfer, an enhanced mixing, and an acceleration of ocean currents following sea ice reduction (sections 2.1, 2.2, and 2.5), aligns with well established expectations and is not particularly surprising. However, the proposed model-based quantification of changes, including the seasonal variability assessment provided in section 2.3, offers some insights.

I also have some reservation about the study's approach to establishing an empirical relationship between ocean surface stress and wind speed (section 2.4 and Figure 5). From what I understand, this relationship does not take into account ice concentration and thickness, crucial variables influencing sea-ice internal stress, and hence the "dampening factor" introduced by the study. Given the study's focus on the effects of reduced sea ice on this dampening, excluding these variables from the analysis is questionable. I expect this omission to result in a large uncertainty for the empirical relationships shown in Figure 5, an aspect not addressed in the paper.

Reviewer #3 (Remarks to the Author):

This study, "Future sea ice weakening amplifies wind-driven Arctic Ocean spin-up", provides an important look at how the transfer of momentum from the atmosphere into the ocean in the Arctic is likely to change in future climate scenarios. This momentum transfer is a central control on future conditions in the upper Arctic Ocean, but studies to date have only conjectured or extrapolated about future changes to ocean surface stress. Assessment of those changes across an ensemble of climate models provides a novel set of results that will aid in our understanding of future Arctic change, and is fitting for the audience of Nature Communications.

Except for a few points (detailed in my "Major Comments", below), I am generally satisfied with the analysis and methods used in the paper. However, some analysis is unfortunately hampered by the lack of consistently available stress information in CMIP model output, forcing the authors to make some extrapolations that they don't necessarily justify (specifically: is NorESM2-MM representative of other models?). Of greater concern, there are some key challenges associated with the interpretation of CMIP-class model projections of surface stress. In particular, past studies have shown that even the sign (increasing or decreasing) of calculated surface stress trends are sensitive to details of surface stress parameterization (Martin et al., 2016; Sterlin et al., 2023). The authors do acknowledge this directly, and make some attempts to investigate the impact in §2.4 of the manuscript, but at this point, I am not convinced that the issue is resolved. I expand on this in my "Major comments". Given the sensitivity of these trends to surface parameterizations, it is surprising that no details are given in the manuscript regarding the parameterization schemes used in different models in the ensemble, and thus there is a lack of important contextual information needed to fully assess the results of the study.

A central question arising from the result that ocean surface stress is increasing is whether that increase is driven by changing wind conditions, changing ice cover, or changing ice properties (i.e., mechanical strength). The title of the paper puts forward the central thesis: that it is the weakening of the sea ice that is the most important driver. This is based on analysis in sections 2.2 and 2.3 of the paper. However, the figures and analysis presented in those sections could do a better job of making that point (and in particular, in separating the impact of changing ice cover and changing ice strength — the two are related, but could have separate effects).

Despite these limitations, the study is of generally high quality and interesting, and the writing and presentation are mostly clear (though could benefit from some reorganization, and some points could be stronger). I think that it will likely warrant publication after some changes to address the concerns listed below.

Major comments:

- The title of the manuscript, while a technically correct interpretation of the results, makes it sound like the paper will focus on the "spin-up", rather than on the increase of ocean surface stress (which *could* result in a spin-up). I wonder if an alternative title could better communicate the main results.

- The authors consistently use the NorESM2-MM model as a case study throughout the manuscript, as out of the models assessed, only it provides a detailed enough set of outputs to analyze the breakdown of changes in the surface stress and the ocean mixing. However, there is no discussion in the paper of how representative this particular model might be of the rest of the ensemble (both in terms of stress trends, but also other processes — e.g., sea ice loss). The authors should provide this information explicitly either in the main body of the manuscript or within the "Methods" section (e.g., maybe including separate lines showing NorESM2-MM results in Figure 2a,b, and others). It would be further useful if the idea of NorESM2-MM as a case study was introduced early in the manuscript, for example in the final paragraph of the introduction section. Many of the results in the manuscript lean on an assumption that analyzing NorESM2-MM gives a good view of how the full ensemble behaves, but there is limited evidence provided that that assumption is valid.

- Section 2.2: Despite presenting one of the primary sets of results from the paper, interpretation of this section and parsing of those results is difficult. In particular, by looking at the trends of τ_{io} and $(1-A)\tau_{ao}$ it becomes difficult to understand whether contributions arise from stress changes or ice concentration changes. Figure 3b attempts to separate these effects, but as that only looks at one part of the equation, it is not sufficient. Moreover, the label "ice-ocean stress" is applied to the full term τ_{io} in this section, which means that in other parts of the manuscript, it is unclear if "ice-ocean stress" means τ_{io} or τ_{io} (and similarly for other terms). In addition, there is some confusion in the sign convention in Figure 3c; I suspect that the positive valued trends in AF_i are meant to indicate that the resulting trend in total stress from changes in AF_i is positive, but when I look at the figure it appears that there is a positive trend in AF_i itself (i.e., the internal stress is *increasing* — note that that interpretation is consistent with the legend). This took me some effort to parse on my first read-through. There are probably a range of strategies that might make the overall set of results in this section clearer, and I leave it to the authors to decide for themselves. Personally, I would suggest some reorganization of this section together with the following section (§2.3) including changing the order of figures 3 and 4. I would also introduce the total atmosphere surface stress $\tau_a = (1-A)\tau_{ao} + A\tau_{ai}$ at the same time as introducing τ_o and discussing them together (to help separate wind versus ice effects). I might also consider presenting trends of mean τ_{ao} and τ_{io} instead of integrated $(1-A)\tau_{ao}$ and $A\tau_{io}$, thereby separating the change in ice concentration.

- Section 2.4: This section attempts to contend with some of the challenges associated with different surface roughness parameterizations (as mentioned in my summary comments, above). Here, I have a few concerns:

(1) In both Martin et al. (2016) and Sterlin et al. (2023), the inclusion of variable drag parameterizations causes the tendency in modelled ocean surface stress to change sign and have a general *decreasing* trend. In this section of the manuscript, the authors suggest that these effects will imprint in different ways on the different models, but that it is not a concern because there is nonetheless a robust increase of ocean surface stress across all these models (e.g., as seen in Figure 2a). However, there is no discussion about the different surface roughness parameterizations used in these models. To my knowledge (which may be incomplete), parameterizations of variable sea ice drag coefficients are only available in CICE (Tsamados et al., 2014) and NEMO-LIM3 (Sterlin et al., 2023) and it's not clear to me if those parameterizations would

have been enabled for the model runs included in the present study. I suspect that the model ensemble here shows a consistent increasing stress trend because they all (or most) use constant drag coefficients. To clarify this, the authors should, at minimum, indicate in Table 1 (or elsewhere) which of the models use variable neutral drag coefficients, and also which include corrections for non-neutral drag in either the atmosphere or ocean, and provide more discussion of this context.

(2) In addition to the above, there is still significant uncertainty about the accuracy of extant drag parameterizations (e.g., Brenner et al., 2021; Sterlin et al., 2023; Bateson 2021 [<https://doi.org/10.48683/1926.00098821>]; Bateson et al, 2022 [doi.org/10.5194/tc-16-2565-2022]), whether applied or not, which would imprint as uncertainty on future projects of surface stress.

(3) The authors indicate that the empirical relationships between wind speed and stress (in Figure 5) can help further elucidate the impacts of the different stress parameterizations across models. While the analysis does highlight variability between the models, the point of this analysis is not clearly made by the authors. As such, my opinion is that it does not currently add much value to the manuscript. Additionally, as with the previous points about the representativeness of NorESM2-MM, it is not clear how representative the 5 models in the section are of the full ensemble (they were chosen because of the output variables available, and not because they span the set of ensemble results). Furthermore, the description of the methods used to establish empirical relationships (in §4.3 #5,6) is lacking detail. In particular, statistics/uncertainty of the empirical fits should be assessed (to me, it seems that these fits could change with time or region, so I am not convinced of their robustness). The authors might consider a supporting figure showing an example of one of these fits with fully scattered data and uncertainty ranges.

(4) In contrast, the damping factor presented in this section seems like a powerful and useful tool (and it sounds like it could be calculated for more than just those 5 models, as it doesn't rely on τ_{ao} , τ_{ai} , τ_{io} separately, but just bulk τ_a and τ_o). I think more could be done with this. Note that $\tau_a - \tau_o$ provides some measure of F_i , even in models that don't explicitly report it (see equation 2 in Martin et al., 2014).

- Figure presentation: Many figures use a repeating colour pattern with green/teal lines and orange lines. Repetition of colour consistently across figures can greatly aid readers, so I appreciate this. While that repetition has been done to some extent here, the meanings of the colours vary somewhat throughout the manuscript. In Figures 2 and 6, green/orange correspond to fall/winter; in Figures 3 and 5 they correspond to atmosphere-ocean/ice-ocean stresses, and in Figure 4 they refer to past/future conditions. It might improve readability if each category pair was given a distinct colour pair. Also, is there a reason that the doughnut colour for the "damping factor" in Figure 5 matches the teal colour used for τ_{ao} ? Maybe choose something entirely different as it is not part of one of the category pairs.

- It seems that changes in sea ice drift speeds would also be an important aspect of this story: a reduction in sea ice internal stress would likely lead to a larger proportion of the ice being in "free drift", and being more mobile. I'm surprised that the authors did not include any assessment of ice velocity changes.

- One of the main ideas in Martin et al, (2014, and also explored by others) is that there is some intermediate ice concentration at which the net ocean surface stress is maximized due to in

interplay in differences between sea ice and open-water roughness, and sea ice internal stress. While that background obviously guides some aspects of the present study, this concept wasn't explored directly here and it seems that there is an opportunity to include those ideas more explicitly. In particular, I wonder if the future increase in ocean surface stress is related to a change in the shape of the τ_o versus A curves at high A (because of mechanical ice weakening), or if the curves stay the same, but future conditions have a greater proportion of time at lower ice concentration conditions (near the intermediate value of A that would maximize stress input). Is the shift in surface stress trends after 2060 for Summer and Fall in Figure 2a a reflection of shifting ice conditions *past* this maximum and into a less effective open-water regime?

Other minor comments:

- Fig 2: Could benefit from a panel (like a,b) showing sea ice area trends across the ensemble for the different seasons
- Fig 2: Could benefit from an additional panel like (c) showing the stress, but for future 2085-2100 conditions.
- L124: Any speculation about why the Fall winds speed curve flattens after 2060? Or why the Summer wind speed increases through the full period while the Summer ocean surface stress curve also flattens after 2060?
- L133-135: The phrasing here makes the cause and effect sound backwards. Maybe instead of "...we can anticipate comparable trends in surface wind speed." you could say "...we can anticipate these trends follow changes in surface wind speed." However, I think a re-organization of these ideas would be better. Especially if you include sea ice area trends in Figure 2, then you could start with sea ice change, then wind speed change, then stress change (which is a combo of both effects), and thus build to the analysis of "which one was it?".
- L138: As with the previous comment, this is awkward phrasing.
- L185-187 and Fig 2: In fact, the Winter stress trend matches what might be predicted by the wind speed change pretty well (Winter wind speed trend +18%, and quadratic drag gives $1.18^2=1.39$ or a +39% stress increase, consistent with Winter stress trend +38% in Figure 2a). It's the Summer and Fall trends that are weird: why is stress increasing more slowly than wind speed during those seasons? Please provide some commentary on this (including the expectations based only on wind speed changes).
- L205: Awkward phrasing — the ocean model is forced by τ_o (which is computed from τ_{ao} and τ_{io}).
- L208: Introduce NorESM2-MM as a "case study" in the introduction, and somewhere explain how representative it is.
- L218-219: Be more specific here. Stress out of the atmosphere is usually thought to be greater over ice than open water, which contrasts with the statement in the manuscript. This suggests that internal stresses are also included in that statement, but it isn't clear.
- L279-282: This statement could be made more clear. Is the ice weaker because it has a lower concentration (on average), and strength is a function of concentration (and thickness)? Or is internal stress contribution less because the lower concentration means that there is less ice available to contribute to internal stress (integrating over a smaller area)? Are the results presented

sufficient to allow us to separate those effects?

- L290–293: While the increase in stress is robust across models, it's not clear that there is sufficient evidence to say that the decomposition is robust across models (it probably is, but the results don't necessarily prove it).

- L297–301: This feels more like content for the introduction rather than the results.

- Figure 4: I don't think the 2055-2070 mean adds much to the figure and it isn't discussed in the text (side note: this is a great figure)

- L363–364: Which previous studies suggest a shift in the seasonality of Arctic surface stress? Also, what would be the impact of this shift?

- Figure 5: Fonts in the legend are small/hard to read.

- L421/§4.3 #7: Does normalization by τ_a in the "damping factor" lead to issues when τ_a is a small value?

- Section 2.5: Much of this content feels like background that would be more appropriate in an introduction section instead of a results section (e.g., L451-459 and especially L499-513). It seems that more could be said in this section (especially about shifting patterns of ocean surface velocity or kinetic energy), especially given the reference to these changes in the manuscript's title.

- L460/L495: Kinetic energy is not an appropriate proxy for turbulent mixing, which will depend on a balance of both kinetic and potential energy. (It is still appropriate to discuss kinetic energy in the context of mixing)

- L499-522: It isn't convincing why the Beaufort Gyre in particular is singled out for evaluation when there are a range of climatically important areas that could equally be considered. The sensitivity of some of the results (e.g., FWC) to other, likely more important factors (as described in L516–518), makes these results feel fairly contrived. If the authors plan to keep an evaluation of the Beaufort Gyre in future revisions, they should frame the analysis in terms of the "ice-ocean governor" effect (e.g., Meneghello et al., 2018) that they described in the introduction (I suspect that was the motivation in this evaluation, but it isn't linked back to those ideas).

- L547: Variability in bottom roughness should also cite Brenner et al., 2021 and Cole et al., 2017 (as already cited elsewhere in the manuscript).

- L647: "All trends presented are statistically significant unless otherwise stated" At what confidence level?

- There are a few entries in the references with missing fields (indicated as "???"), and at least one with an error in the title (Castellani et al., 2014)

Dear Reviewers,

Thank you for the thorough and constructive reviews of our paper. We greatly appreciate the effort you have taken to delve into all the details, and we believe your suggestions have significantly enhanced the manuscript. We have addressed all comments and updated the manuscript accordingly. Our detailed, point-by-point responses are provided below in blue font. Additionally, a tracked-changes document is attached for your reference.

Reviewer #1 (Remarks to the Author):

Review of “Future sea ice weakening amplifies wind-driven Arctic Ocean spin-up ” by Muilwijk et al.

In this manuscript Muilwijk and co-authors present an analysis of future changes in the Arctic. The reduction in sea ice concentration leads to an increase in the surface stress experienced by the ocean, with consequences for surface speed, freshwater storage, and diapycnal mixing. The analysis is thorough, the results are clear, and the conclusions are robust. This manuscript will make a valuable contribution to the literature. I have a few minor comments that I would like the authors to address prior to publication.

Edward Doddridge

General Comments

Summer wind speed vs stress

The summer wind speed increases linearly through time out to 2100, but the wind stress plateaus. Is there an obvious explanation for this decoupling? Is it related to the quadratic drag term in the wind stress – perhaps the wind speed is only increasing in the places with relatively slow winds, and therefore makes a negligible contribution to the total stress? I understand that this manuscript is focused on the fall and winter seasons, but this discrepancy is odd, and, in my opinion, it warrants mentioning.

Thank you for raising this point. We agree that this warrants mentioning and did some further analysis, from which we conclude that the counter-intuitive plateauing of ocean surface stress in summer (Fig. 2a) while wind speed keeps increasing (Fig. 2b) is possibly due to the decreasing sea ice concentration and total ice coverage. As discussed in the revised version of our manuscript (see new supplementary Figure S4 and sections 2.4 and 2.5) most models indicate a maximum (some more narrow, others wider) in ocean surface stress for ice concentration between 10% and 90%. As sea ice coverage decreases over the course of the century, ocean surface stress will first increase and later decrease as basin-mean ice concentration sinks below this maximum (as argued by Martin et al. (2014)). However, we find a plateau rather than a peak because the steadily increasing wind speed could mask the decreasing effectiveness of the partially ice-covered ocean surface to take up the momentum. Additional effects, such as spatial distribution of wind peaks, atmosphere boundary layer stability, and the averaging across various models may play a role as well. For example, from the new Figure 3d (note this shows fall instead of summer; the plateauing addressed is also visible for fall, new Fig. 2a) we see that not all models show a clear plateauing. The following explanation has been added to the manuscript:

“Additionally, in summer and fall, wind speed increases linearly through 2100, but surface stress plateaus (Fig. \ref{fig:intro}a-b). We do not investigate the plateauing in detail but suspect it is related to a change in average sea ice concentration. As discussed in Section 2.5, most models suggest a maximum ocean surface stress for ice concentrations between 10\% and 90\% (for grid cells with more

than 0% ice). Stress increases initially with decreasing ice concentration, but later decreases as basin-mean ice concentration reduces to sub-optimal conditions (as argued by \cite{martin2014seasonality}). Consequently, continuously increasing wind speed results combined with reduced sea ice concentration can result in a plateau in wind stress. Additionally, the plateauing of the wind stress time series may partly be an artifact of the multi-model averaging, as individual models display a less coherent time evolution (Fig. \ref{fig:spaghetti}).”

Sea ice internal stress

The diagnostics shown in the manuscript highlight the role of internal ice stress. Perhaps it is a misguided notion, but I had always considered that much of the stress that the ice pack absorbs is transferred horizontally into the land boundaries surrounding the ocean. I would have appreciated a bit more discussion about what happens to the stress that is absorbed by the ice pack.

Thank you for this question. Most of the internal stress is eventually converted into sea ice deformation or kinetic energy dissipation over time. The detailed implementation of this process varies across different models, but in general, the sea ice internal stress is a measure of ice strength to resist such deformation and typically acts to reduce the motion caused by the wind forcing. From a continuum perspective as used in the models discussed here, low ice concentration ($A < 70\%$) enables the mix of differently sized floes to move freely without deformation. This is termed free drift and enables momentum transfer with much less dissipation. With higher concentrations, floe-floe interaction is considered to restrict the movement of the ice pack and part of the wind energy is transferred into internal stress (eventually converting kinetic energy into heat, with varying representation in models in use). The underlying assumption is that the ice pack possesses plastic and elastic behavior, and it remains stationary until stress surpasses a threshold (Hibler, 1970). If ice is pushed together hard enough the ice will eventually break and pile up into ridges (sea ice deformation - which is an irreversible process). The relationship between internal stress and sea ice deformation is termed sea ice rheology, and it depends on ice compactness, ice strength, strain, and strain rate (Leppäranta and Omstedt, 1990). When ice is pulled apart, internal stress may also be released and converted into kinetic energy (this is only possible in elastic rheology). The sea ice rheology is incorporated into the sea ice momentum equation through the internal stress tensor, denoted as σ :

$$m\left(\frac{D\vec{u}}{Dt} + f\hat{k} \times \vec{u}\right) = A(\vec{\tau}_{ai} + \vec{\tau}_{io}) + mg\nabla H + \nabla \cdot \underline{\underline{\sigma}}$$

Ice dynamics exhibit a wide range of rheological characteristics (Heorton et al., 2018), making sea ice rheology modeling complicated. Various approaches exist, including viscous-plastic, elastic anisotropic plastic, and brittle models. Consequently, the internal stress tensor σ , is the most uncertain term in the sea momentum equation. A compressed version of the above explanation has been included in the introduction chapter. Additionally, we have included some discussion on what happens to internal stress in chapter 2.2 and in the conclusion.

Specific comments

Line 127: “more significant” is confusing in this context. You’ve just described statistically significant trends, and I think what you mean here is ‘more substantial’ or ‘larger’. I suggest changing the wording. We agree the wording is confusing. This has been changed to “more substantial” in Line 127.

Lines 153-173, Figures 2c) and 2d): There’s a lot of unused space on these panels – do we need to see so much land? A polar stereographic projection might focus the readers’ attention on the Arctic ocean and take up less space.

Agreed. We have replaced these with polar stereographic projection maps.

Line 170, Figure 2d): These figures are beautiful, but I recommend switching out the diverging colourmap in 2d) for a one sided colourmap like you've used in 2c). As far as I can tell, there is nothing particularly special about the value 0.27 N/m^2 , so highlighting areas that are higher and lower than this value isn't useful.

Good suggestion. We have replaced the colormap with a one-sided map.

Line 284: "Consequently, the reduction in internal sea ice stress (red bars, Fig. 3c) contributes to a positive trend in ocean stress throughout all months (gray bars, Fig. 3a)."

I suspect this is a linguistic issue when trying to describe a reduction in magnitude of a negative term, but the above sentence does not make sense to me. The red bars in Figure 3c) show a positive trend (increase) in the total internal stress, not a reduction.

Thank you for this good point. Yes, this is an issue of describing an increase in the magnitude of a negative term. The other reviewers also pointed to this section as somewhat confusing. We have extensively restructured and reformulated the entire text to enhance clarity. To avoid further confusion with the sign of "an increase of a negative term", we have included an extra panel illustrating the trend in internal stress (negative) and relabeled the decomposed trends in the subsequent figure as "contribution from atmosphere-ice stress" and "contribution from internal stress." We believe these modifications will significantly enhance the clarity and consistency of the section, making it easier for readers to follow.

Lines 388-403, Figure 5: There's a lot going on here. Well done on presenting the information so succinctly. It was not immediately obvious to me that two of the models showed a negative dampening effect – perhaps it would be helpful to plot the doughnuts, or the text labels, in a different colour for those two models. The key in the bottom right is helpful, but needs some work. 't_a' isn't described anywhere in the figure or the caption, and the 'clockwise climate response' label is plotted next to a doughnut that shows a counter-clockwise response (i.e., more dampening). (Actually, reading further in the text, I think this bit of the key is actually meant to indicate the change between two doughnuts – that wasn't clear to me when looking at the figure.)

Thank you for these suggestions. This figure (now Fig. 6) has been extensively revised based on enhanced analysis in this section. This section now delves deeper into the differences in model parameterizations, providing a more comprehensive understanding of the role of ice. We have calculated the doughnut charts for all models, placing particular emphasis on the transition from past to future, as recommended. Additionally, we have conducted new analysis on the relationship between stress and sea ice concentration. During the refinement of our analysis on the dampening factor, we identified and addressed an issue in the code related to grid conversions between atmosphere and ocean grids. Subsequently, the doughnut charts have been updated accordingly. With this correction, it becomes evident that all models exhibit a dampening effect, aligning more closely with expected outcomes and enhancing the consistency of our findings. The other reviewers raised concerns regarding the statistical significance of the empirical relationship presented in the earlier version of the figure, and whether the 5 models showcased were representative of the entire model ensemble. As a result, this analysis has been omitted from the revised version.

Lines 427-8: Perhaps this result could be shown on an additional panel in Figure 5? It takes a lot of interpretation to glean it from the panels presented.

We agree that there was a lot going on in the original Fig.5. In the updated figure (now Fig. 6), emphasis is placed on the doughnut plots, highlighting the consistent climate response as the focal point. Additionally, the magnitude of the climate response is clearly depicted for each model.

Lines 461-471: Figure 6a-c): These plot speed (a scalar), not velocity (a vector). Current speed
Agreed, and thanks for noticing this detail. All figure legends and text throughout the manuscript have been updated accordingly.

Lines 521-522: “This further supports the impact of reduced energy loss to internal stress.” It’s not clear to me how this sentence follows the previous one. Some extra guidance would be helpful.

Agreed. This has been expanded to: “The effect of enhanced Ekman pumping is limited to the winter season (+33%), whereas a negative trend in Ekman pumping is projected by the multimodel mean for fall (-26%). This seasonal variation is likely due to winter exhibiting the largest trend in surface stress. Given that the latter is related to reduced energy loss to internal stress (as discussed in Section 2.2), we conclude Ekman pumping to be affected indirectly as well.”

Lines 571-574: Should this instead say “... differentiate between ice-ocean and atmosphere-ice stress”? Thank you for spotting this error. Changed to “atmosphere-ice stress in line 571”

Lines 583-585: It is almost certainly true that internal variability is dwarfed by the forced trends, but it would add weight to your argument if you presented one time-series figure from a low emissions pathway in the supplementary material.

We are not convinced that including a time-series figure from a low emissions pathway in the supplementary material would add weight to our argument. It is widely understood and anticipated that under a low emission scenario, internal variability will play a more prominent role and consequently scenario runs exhibit a smaller forced trend. Given this understanding, and the fact that our paper focuses on a mechanistic understanding of a changing Arctic, and not the differences between different emission scenarios, we believe that including such a figure would not substantially contribute to our narrative. Moreover, generating additional data analysis for this purpose would require a significant amount of time and computational resources. Therefore, we maintain that the omission of this figure does not detract from the clarity or validity of our findings. However, the following reference has been added to strengthen this argument: “No other emission scenarios are included because the focus of this study is the mechanistic understanding of a changing Arctic, not the differences between different scenarios, and we thus seek the best available signal-to-noise ratio. Furthermore, for winter (which is the primary focus season of our study), more than 85% of the projection uncertainty is due to model differences, not the emission scenario (see Figure 1d in Bonan et al., 2021).”

<https://doi.org/10.1088/1748-9326/abe0ec>

Line 672: I assume “love” is meant to be “over” or something similar.

Correct. Although I do love open water, this is a spelling error and has been corrected to “over”. Thank you for spotting this. Changed in Line 672.

Lines 721-723: The use of parentheses to denote the negative version of a sentence is convenient (difficult to interpret and frustrating) for authors (readers). I strongly suggest rewriting this sentence so that it clearly presents each idea separately and is easier for your readers to understand.

Again, a very good point. Convenient to write, but frustrating to read. The sentence has been rewritten to the following: “If the atmospheric wind stress exceeds the ocean surface stress, the dampening factor will be positive; it is <0% in case the ocean stress is greater. This means that sea ice dampens momentum transfer when the factor exceeds 0%, whereas ice amplifies momentum transfer in case of a negative factor.” Changed in Line 721.

Line 1221, Figure S2 caption: “However, during winter, the positive trend of internal stress reduction (orange line, Fig. d) outweighs the contribution of changing atmosphere-ice stress.” Two things: The total internal stress is the red line, not the orange one; and ‘positive trend of internal stress reduction’ is very convoluted. I recommend replacing with a simpler statement like ‘the reduction in total internal stress’.

Thank you for spotting this error. Orange has been replaced with red and the statement has been simplified as suggested: “However, during winter, the reduction in total internal stress (red line, Fig. d) outweighs the contribution of changing atmosphere-ice stress.” Changed in Line 1221.

Reviewer #2 (Remarks to the Author):

This paper examines the impact of projected future changes in sea ice on surface ocean stress and the large-scale circulation of the Arctic Ocean. It draws on a suite of 16 CMIP6 runs based on a single emission scenario for part of its analysis, and utilizes the NorESM2-MM model for a more detailed analysis of the central component of the study: ice internal stresses. The study finds that diminishing ice cover will lead to increased momentum transfer from the atmosphere to the ocean, accelerating large-scale circulation and increasing turbulent mixing. While the paper also touches on potential effects on biogeochemistry, heat transport and the Atlantic Meridional Overturning Circulation (AMOC), these discussions remain speculative, and the paper does not concretely address these issues.

My major concern is that the main finding, an increase in momentum transfer, an enhanced mixing, and an acceleration of ocean currents following sea ice reduction (sections 2.1, 2.2, and 2.5), aligns with well established expectations and is not particularly surprising. However, the proposed model-based quantification of changes, including the seasonal variability assessment provided in section 2.3, offers some insights.

Thank you for these comments. We appreciate your acknowledgment that our study confirms well-established expectations regarding the increase in momentum transfer, enhanced mixing, and acceleration of ocean currents following sea ice reduction. While these findings may not be surprising, we believe they provide valuable evidence to what up to now only has been speculated on. To our knowledge, none of these points have actually been shown in a model-based framework. Increased transfer was speculated but not proven. We provide a novel and detailed assessment of both the trends and seasonal variability in an ensemble of state-of-the-art climate models, and by doing so highlight the significance of different ice conditions and parameterizations. We acknowledge, however, that despite the robustness of our main findings, there still exists significant uncertainty regarding the parameterization of ice/ocean drag. This uncertainty, e.g. the inclusion of form drag of models could potentially affect the magnitude of the trends. Therefore, we hope that, in addition to offering robust insights into the direction of Arctic trends, we are able to enhance awareness of a process in the Arctic climate system that is currently heavily oversimplified in models but has potentially significant implications.

I also have some reservation about the study's approach to establishing an empirical relationship between ocean surface stress and wind speed (section 2.4 and Figure 5). From what I understand, this relationship does not take into account ice concentration and thickness, crucial variables influencing sea-ice internal stress, and hence the "dampening factor" introduced by the study. Given the study's focus on the effects of reduced sea ice on this dampening, excluding these variables from the analysis is questionable. I expect this omission to result in a large uncertainty for the empirical relationships shown in Figure 5, an aspect not addressed in the paper.

Thank you for your comment and for raising your concerns. The concern regarding the empirical relationship was also shared by other reviewers. Given the uncertain statistical foundation for the empirical relationship and the fact that 5 models might not be representative for the whole ensemble, we have omitted parts of this analysis and refocused on the "dampening factor," which has been refined and calculated for all models. Additional discussion has been added regarding the role of different parameterizations (new section 2.4). This section and the figures have hence been substantially updated, and two new figures have been added. The revised version of the manuscript now includes additional analysis and discussion on how the dampening factor relates to sea ice area, concentration, and thickness as was suggested by the reviewer. For example, we have included details about the different models' sea ice total area and relate this to the strong or weak dampening factor (new section 2.4). Furthermore, a new figure S4 provides additional insight on the relationship between sea ice concentration and maximum stress for all individual models, similar to what was done by Martin et al. in 2014. Additionally, we have included analysis of sea ice thickness changes across the suite of models and include how the change in stress affects sea ice drift speed (new section 2.5). We hope that these

additional analyses address your concerns and contribute to a more comprehensive understanding of the relationship between ocean surface stress and sea ice dynamics.

Reviewer #3 (Remarks to the Author):

This study, "Future sea ice weakening amplifies wind-driven Arctic Ocean spin-up", provides an important look at how the transfer of momentum from the atmosphere into the ocean in the Arctic is likely to change in future climate scenarios. This momentum transfer is a central control on future conditions in the upper Arctic Ocean, but studies to date have only conjectured or extrapolated about future changes to ocean surface stress. Assessment of those changes across an ensemble of climate models provides a novel set of results that will aid in our understanding of future Arctic change, and is fitting for the audience of Nature Communications.

Except for a few points (detailed in my "Major Comments", below), I am generally satisfied with the analysis and methods used in the paper. However, some analysis is unfortunately hampered by the lack of consistently available stress information in CMIP model output, forcing the authors to make some extrapolations that they don't necessarily justify (specifically: is NorESM2-MM representative of other models?). Of greater concern, there are some key challenges associated with the interpretation of CMIP-class model projections of surface stress. In particular, past studies have shown that even the sign (increasing or decreasing) of calculated surface stress trends are sensitive to details of surface stress parameterization (Martin et al., 2016; Sterlin et al., 2023). The authors do acknowledge this directly, and make some attempts to investigate the impact in §2.4 of the manuscript, but at this point, I am not convinced that the issue is resolved. I expand on this in my "Major comments". Given the sensitivity of these trends to surface parameterizations, it is surprising that no details are given in the manuscript regarding the parameterization schemes used in different models in the ensemble, and thus there is a lack of important contextual information needed to fully assess the results of the study.

Thank you for these very constructive and kind comments. We are glad you find our findings interesting and fitting for the journal audience. We acknowledge that the issue concerning the impact of different parameterizations and model variances was not adequately resolved in the initial version of the manuscript. In response to the reviewer's suggestion, we completely restructured Section 2.4 to delve deeper into this aspect and have provided more comprehensive detail. For instance, we have refined and included the dampening factor for all models, along with a more detailed analysis and discussion on the relationship between stress and various sea ice parameters such as area, concentration, and thickness (new Figure 3, Figure 6, Figure 7 and Figure S4), as was suggested by another reviewer. Additionally, we have reached out to all the different modeling centers to obtain details about their parameterizations, which were not accessible from the various model description papers. This information has now been incorporated as well. Moreover, as suggested, we have conducted an additional analysis on the relationship between sea ice concentration and peak stress following Martin et al. (2014). We elaborate on the specifics of these changes in the "Major comments" below. We trust that these expanded sections and supplementary analyses effectively address your concerns and contribute to a deeper understanding of surface stress parameterization in CMIP6 models.

A central question arising from the result that ocean surface stress is increasing is whether that increase is driven by changing wind conditions, changing ice cover, or changing ice properties (i.e., mechanical strength). The title of the paper puts forward the central thesis: that it is the weakening of the sea ice that is the most important driver. This is based on analysis in sections 2.2 and 2.3 of the paper. However, the figures and analysis presented in those sections could do a better job of making that point (and in particular, in separating the impact of changing ice cover and changing ice strength — the two are related, but could have separate effects).

Agreed. We have clarified in the abstract, results and conclusions that: "increase in future surface stress, is driven primarily by a combination of increased surface wind speed and reduced sea ice concentration. A weaker ice pack further amplifies these trends." Additionally the following has been added to the

results and conclusion: “Fully separating the effects of changing ice cover, ice strength, and winds in all individual models exceeds the format of this publication. In NorESM2-MM we found a representative example where changing ice area and wind speed are the primary drivers of surface stress changes. In winter, the reduction in internal stress explains 29-50% of the trend.”

Despite these limitations, the study is of generally high quality and interesting, and the writing and presentation are mostly clear (though could benefit from some reorganization, and some points could be stronger). I think that it will likely warrant publication after some changes to address the concerns listed below.

Major comments:

- The title of the manuscript, while a technically correct interpretation of the results, makes it sound like the paper will focus on the "spin-up", rather than on the increase of ocean surface stress (which *could* result in a spin-up). I wonder if an alternative title could better communicate the main results.

Original: Future sea ice weakening amplifies wind-driven Arctic Ocean spin-up.

We agree and have suggested the following title to add emphasis that the main focus of the paper is surface stress: Future sea ice weakening amplifies wind-driven trends in surface stress and Arctic Ocean spin-up.

- The authors consistently use the NorESM2-MM model as a case study throughout the manuscript, as out of the models assessed, only it provides a detailed enough set of outputs to analyze the breakdown of changes in the surface stress and the ocean mixing. However, there is no discussion in the paper of how representative this particular model might be of the rest of the ensemble (both in terms of stress trends, but also other processes — e.g., sea ice loss). The authors should provide this information explicitly either in the main body of the manuscript or within the "Methods" section (e.g., maybe including separate lines showing NorESM2-MM results in Figure 2a,b, and others). It would be further useful if the idea of NorESM2-MM as a case study was introduced early in the manuscript for example in the final paragraph of the introduction section. Many of the results in the manuscript lean on an assumption that analyzing NorESM2-MM gives a good view of how the full ensemble behaves, but there is limited evidence provided that that assumption is valid.

This is a very good point and we thank you for this suggestion. First, we have introduced NorESM2-MM as a case study at the end of the introduction section as suggested: “One of the CMIP6 models, NorESM2-MM \citep{Seland2020}, is utilized for a more detailed analysis as it provides detailed enough output for a more precise breakdown in stress components and ocean mixing, offering a case study of the processes at play.” We have also added a new figure (now Fig. 3, added here below) showing a spaghetti plot of all the model’s time series of surface stress, sea ice area, and wind speed. This does not only show how representable NorESM2-MM is, but also a more detailed comparison of the various models’ behavior, which is not evident from the collapsed multimodal-mean, but which we think can be useful for many readers. We believe this additional figure strengthens the analysis and also provides insight into the question about the “flattening of the wind speed” brought up by reviewer 1. A detailed discussion on the differences in the model’s mean state, how the mean state affects our findings, and the representativeness of NorESM2-MM has been added to the end of Section 2.1.

- Section 2.2: Despite presenting one of the primary sets of results from the paper, interpretation of this section and parsing of those results is difficult. In particular, by looking at the trends of $A\tau_{io}$ and $(1-A)\tau_{ao}$ it becomes difficult to understand whether contributions arise from stress changes or ice concentration changes. Figure 3b attempts to separate these effects, but as that only looks at one part of the equation, it is not sufficient. Moreover, the label "ice-ocean stress" is applied to the full term $A\tau_{io}$ in this section, which means that in other parts of the manuscript, it is unclear if "ice-ocean stress" means τ_{io} or $A\tau_{io}$ (and similarly for other terms). In addition, there is some confusion in the sign convention in Figure 3c; I suspect that the positive valued trends in AF_i are meant to indicate that the resulting trend in total stress from changes in AF_i is positive, but when I look at the figure it appears that there is a positive trend in AF_i itself (i.e., the internal stress is *increasing* — note that that interpretation is consistent with the legend). This took me some effort to parse on my first read-through. There are probably a range of strategies that might make the overall set of results in this section clearer, and I leave it to the authors to decide for themselves. Personally, I would suggest some reorganization of this section together with the following section (§2.3) including changing the order of figures 3 and 4. I would also introduce the total atmosphere surface stress $\tau_a = (1-A)\tau_{ao} + A\tau_{ai}$ at the same time as introducing τ_o and discussing them together (to help separate wind versus ice effects). I might also consider presenting trends of mean τ_{ao} and τ_{io} instead of integrated $(1-A)\tau_{ao}$ and $A\tau_{io}$, thereby separating the change in ice concentration.

Thank you, we appreciate these suggestions and comments. This section is indeed central to our analysis but we acknowledge that its clarity can be improved as was also pointed out by another reviewer. Following your recommendations, we have extensively restructured and reformulated the entire text to enhance clarity. While we have maintained the original order of sections, we have also revised the previous Figure 3 (now Fig. 4) to improve clarity. We suspect that the main source of confusion in this section is the fact that a reduction in internal stress (negative trend) results in a positive trend in ice-ocean stress. To address this, we have included a panel illustrating the trend in internal stress (negative) and relabeled the decomposed trends in the subsequent figure as "contribution from atmosphere-ice stress" and "contribution from internal stress." We believe these modifications will enhance the clarity and consistency of the section, making it easier for readers to follow.

- Section 2.4: This section attempts to contend with some of the challenges associated with different surface roughness parameterizations (as mentioned in my summary comments, above). Here, I have a few concerns:

(1) In both Martin et al. (2016) and Sterlin et al. (2023), the inclusion of variable drag parameterizations causes the tendency in modelled ocean surface stress to change sign and have a general *decreasing* trend. In this section of the manuscript, the authors suggest that these effects will imprint in different ways on the different models, but that it is not a concern because there is nonetheless a robust increase of ocean surface stress across all these models (e.g., as seen in Figure 2a). However, there is no discussion about the different surface roughness parameterizations used in these models. To my knowledge (which may be incomplete), parameterizations of variable sea ice drag coefficients are only available in CICE (Tsamados et al., 2014) and NEMO-LIM3 (Sterlin et al., 2023) and it's not clear to me if those parameterizations would have been enabled for the model runs included in the present study. I suspect that the model ensemble here shows a consistent increasing stress trend because they all (or most) use constant drag coefficients. To clarify this, the authors should, at minimum, indicate in Table 1 (or elsewhere) which of the models use variable neutral drag coefficients, and also which include corrections for non-neutral drag in either the atmosphere or ocean, and provide more discussion of this context.

Thank you for bringing this up. We completely agree, and we have added a more detailed discussion and new analysis regarding the effect of different parameterizations. We have also been in contact with the individual modeling centers asking for detailed information about the parameterization, which is now included in the table. Section 2.4 has been completely rewritten (following your comments below) and is now called "Uncertainty due to the parameterization of ice-ocean drag". We have also added discussion and speculation on how the findings may be affected by the potential inclusion of form drag following the results of Sterlin et al., (2023) and Martin et al., (2016). The new section includes the following discussion:

"The momentum transfer from the atmosphere to the ocean and sea ice, as well as from sea ice to the ocean, varies across models due to differences in the bulk formulae, specifically in drag coefficients at air-ice and ice-water interfaces \citep{martin2016impact, Sterlin2023}. Furthermore, the transfer representation is simplified, as sea ice roughness is often set to a constant value despite its known large variability, which depends on the floe shape, size, and thickness distribution \citep{tsamados2014impact, castellani2014variability}. While more advanced parameterizations, such as the sea-ice-state-dependent form drag, are available in some models like CICE \citep{tsamados2014impact} and NEMO-LIM3 \citep{Sterlin2023}, all CMIP6-deck simulations used in this study applied a constant ice-ocean drag coefficient (as per personal communication with the various modeling centers). To assess the variability associated with the wide range of drag coefficients (Table \ref{table_models}) and momentum transfer formulations in the models, we introduce the "dampening effect," inspired by the "amplification index" employed by \citep{martin2016impact}.

(...)

While additional diagnostics would be needed to assess whether the partition of the various stress components is consistent across the different models, another large uncertainty lies in the drag formulation itself. However, more advanced parameterizations, e.g. of the ice-state-dependent form drag, that are available in some models \citep{tsamados2014impact}, are still subject to substantial uncertainty \citep{brenner2021comparing, Sterlin2023, Bateson2021, Bateson2022}, with previous studies demonstrating that even the sign (increasing or decreasing) of calculated stress is sensitive to the details of stress parameterization \citep{martin2016impact, Sterlin2023}.

Although more advanced drag formulations are expected to alter the total stress response, we have shown that the primary driver of the trends is the increasing wind speed, which impacts all stress components. Additionally, a future thinning of the ice pack will result in a weakening of the ice. Thus,

despite uncertainties in parameterizations, it is reasonable to conclude that overall Arctic Ocean surface stress will increase due to the combination of increasing winds and reduced internal stress, and the estimates provided by the CMIP6 models represent the best available at this point.”

(2) In addition to the above, there is still significant uncertainty about the accuracy of extant drag parameterizations (e.g., Brenner et al., 2021; Sterlin et al., 2023; Bateson 2021 [https://doi.org/10.48683/1926.00098821]; Bateson et al, 2022 [doi.org/10.5194/tc-16-2565-2022]), whether applied or not, which would imprint as uncertainty on future projects of surface stress. Agreed. This point and the new references have been included to the discussion.

(3) The authors indicate that the empirical relationships between wind speed and stress (in Figure 5) can help further elucidate the impacts of the different stress parameterizations across models. While the analysis does highlight variability between the models, the point of this analysis is not clearly made by the authors. As such, my opinion is that it does not currently add much value to the manuscript. Additionally, as with the previous points about the representativeness of NorESM2-MM, it is not clear how representative the 5 models in the section are of the full ensemble (they were chosen because of the output variables available, and not because they span the set of ensemble results). Furthermore, the description of the methods used to establish empirical relationships (in §4.3 #5,6) is lacking detail. In particular, statistics/uncertainty of the empirical fits should be assessed (to me, it seems that these fits could change with time or region, so I am not convinced of their robustness). The authors might consider a supporting figure showing an example of one of these fits with fully scattered data and uncertainty ranges. This is a very valid point and has also been raised by other reviewers. Considering that the five models might not be representative of the entire ensemble and the relatively weak statistical foundation of the empirical fits, we have chosen to omit this analysis. Instead, we have focused on the more robust metric, the “dampening factor,” which has now been included for all models, as suggested by the reviewer.

(4) In contrast, the damping factor presented in this section seems like a powerful and useful tool (and it sounds like it could be calculated for more than just those 5 models, as it doesn't rely on τ_{ao} , τ_{ai} , τ_{io} separately, but just bulk τ_a and τ_o). I think more could be done with this. Note that $\tau_a - \tau_o$ provides some measure of F_i , even in models that don't explicitly report it (see equation 2 in Martin et al., 2014). We agree and have focused our analysis around this tool. See new figure and text below.

Despite significant inter-model differences, all models indicate an overall damping effect ranging between 8% and 39% (Fig. 6). Notably, CESM2 and GFDL-ESM4 exhibit particularly strong damping effects, while CMCC-ESM2 and MIROC6 show very weak damping effects. Interestingly, there appears to be no correlation between the average damping effect and the drag coefficient (Table 1), suggesting that sea ice state likely plays a crucial role as well. However, the damping effect is also evident in Figure 3, where we observe that CESM2 and GFDL-ESM4 exhibit average wind speeds but particularly low ocean surface stress. In contrast, CMCC-ESM2 and MIROC6 demonstrate particularly high ocean surface stress.

Despite the large variations across models, they all exhibit a consistent climate response (negative), indicating a reduced damping effect of sea ice in the future. Furthermore, the magnitude of this reduction falls within a relatively narrow range of -2% to -18%, underscoring a robust response. The average multimodel mean depicts an initial damping effect of 19% at the beginning of the century, which decreases by 9% towards the end of the century.

In summary, we find that sea ice consistently dampens ocean surface stress in all CMIP6 models, with this effect expected to diminish as sea ice retreats. This supports the findings from the NorESM2-MM case study. The robustness of these findings is evident within the current ensemble of CMIP6 models. However, it remains uncertain whether these conclusions would differ if the models had employed more intricate form drag formulations. To address this uncertainty, separate scenario simulations incorporating these formulations need to be conducted. Nevertheless, as we have shown, increasing wind speed is the primary driver of the trends, and this impacts all stress components. Thus, despite uncertainties in parameterizations that could affect the role of ice and the magnitude of trends, it is reasonable to conclude that overall Arctic Ocean surface stress will increase due to increasing winds. However, the absolute magnitude of this increase remains uncertain, and the estimates provided by the CMIP6 models represent the best available at this point.

- Figure presentation: Many figures use a repeating colour pattern with green/teal lines and orange lines. Repetition of colour consistently across figures can greatly aid readers, so I appreciate this. While that repetition has been done to some extent here, the meanings of the colours vary somewhat throughout the manuscript. In Figures 2 and 6, green/orange correspond to fall/winter; in Figures 3 and 5 they

correspond to atmosphere-ocean/ice-ocean stresses, and in Figure 4 they refer to past/future conditions. It might improve readability if each category pair was given a distinct colour pair. Also, is there a reason that the doughnut colour for the "damping factor" in Figure 5 matches the teal colour used for τ_{ao} ? Maybe choose something entirely different as it is not part of one of the category pairs.

Agreed. We have updated colors in the Figures to ensure that there is no overlap in category pairs. The color pair of orange and green is now only reserved for fall and winter.

- It seems that changes in sea ice drift speeds would also be an important aspect of this story: a reduction in sea ice internal stress would likely lead to a larger proportion of the ice being in "free drift", and being more mobile. I'm surprised that the authors did not include any assessment of ice velocity changes. Indeed, changes in sea ice concentration and wind stress also affect sea ice drift. In response to this consideration, we have included an assessment of sea ice drift speed in the new section 2.5. This section also delves into sea ice thickness and examines the relationship between thickness and stress as was suggested by another reviewer. See new figure below.

- One of the main ideas in Martin et al, (2014, and also explored by others) is that there is some intermediate ice concentration at which the net ocean surface stress is maximized due to interplay in differences between sea ice and open-water roughness, and sea ice internal stress. While that background obviously guides some aspects of the present study, this concept wasn't explored directly here and it seems that there is an opportunity to include those ideas more explicitly. In particular, I wonder if the future increase in ocean surface stress is related to a change in the shape of the τ_{ao} versus A curves at high A (because of mechanical ice weakening), or if the curves stay the same, but future conditions have a greater proportion of time at lower ice concentration conditions (near the intermediate value of A that would maximize stress input). Is the shift in surface stress trends after 2060 for Summer and Fall in Figure 2a a reflection of shifting ice conditions *past* this maximum and into a less effective open-water regime? Thank you for this suggestion. This topic was initially explored for NorESM2-MM but was not included in the manuscript. We have now performed a similar analysis for all

models to investigate whether the net stress is maximized at certain sea ice concentrations. A new Figure S4 has been added as well (see below). From the manuscript: “Martin et al. (2014) explored the relationship between surface stress and sea ice concentration in detail, finding a stress peak at approximately 80% ice concentration. We conducted a similar analysis across all CMIP6 models (Fig. S4). Each model’s sea ice concentration was binned with a width of 2%. Within each bin, ocean surface stress was plotted against sea ice concentration. Additionally, the results were normalized by wind speed to remove the effects of spatial and temporal variations in winds. Our findings are not as definitive as those reported by Martin et al. (2014) and need to be treated with caution. Their analysis was conducted using daily data, whereas our model data is available only as monthly averages. This, combined with the coarse spatial resolution of the models, provides a limited statistical foundation for this analysis. Moreover, over 80% of grid cells (in all models) have sea ice concentrations either above 95% or below 10%, which biases the relationship, as very few grid cells provide information about the empirical relationship between 10% and 95% concentration. Taking this into consideration, our results do not exhibit a clear stress peak at 80%, but they do suggest a maximum ocean surface stress for ice concentrations between 10 and 90%. Since most models show a gradual increase in stress with rising sea ice concentration, followed by a decrease towards 100% ice concentration (consistent with Martin et al. (2014)), it can be hypothesized that if more grid cells had lower ice concentration, the average stress would be lower. While this is visible in some models, there is no clear shift in the normalized relationship from the beginning to the end of the century across the model suite. The key takeaway here aligns with what was found from Figure 6: there are significant inter-model differences regarding the effect of sea ice on moderating momentum transfer.”

Other minor comments:

- Fig 2: Could benefit from a panel (like a,b) showing sea ice area trends across the ensemble for the different seasons. Agreed, this has now been included in the new Figure 3.

- Fig 2: Could benefit from an additional panel like (c) showing the stress, but for future 2085-2100 conditions. This was considered, but the figure is very similar, and instead the “trend” in panel d) was chosen to make the temporal difference more clear. We argue that adding another “mean” panel does not add to the story.

- L124: Any speculation about why the Fall winds speed curve flattens after 2060? Or why the Summer wind speed increases through the full period while the Summer ocean surface stress curve also flattens after 2060? Thanks for this, this question was also raised by Reviewer #1. The counter-intuitive plateauing of ocean surface stress in summer (Fig. 1a) while wind speed keeps increasing (Fig. 1b) is likely due to the decreasing sea ice concentration and overall ice coverage. As discussed in the revised version of our manuscript (see new supplementary Figure S4 and sections 2.4 and 2.5) most models indicate a maximum (some more narrow, others wider) in ocean surface stress for ice concentration between 10 and 90%. As sea ice coverage decreases over the course of the century, ocean surface stress will first increase and later decrease as basin-mean ice concentration passes this maximum (as argued by Martin et al. (2014)). However, we find a plateau rather than a peak because the steadily increasing wind speed is masking the decreasing effectiveness of the partially ice-covered ocean surface to take up the momentum. Additional effects, such as spatial distribution of wind peaks, atmosphere boundary layer stability, and the averaging across various models may play a role as well. For example, from the new Figure 3d we see that not all models show a clear plateauing, and it is thus also partly an artifact from the multi-model averaging . See previous comment to comment from Reviewer #1 for the added explanation.

- L133–135: The phrasing here makes the cause and effect sound backwards. Maybe instead of "...we can anticipate comparable trends in surface wind speed." you could say "...we can anticipate these trends follow changes in surface wind speed." However, I think a re-organization of these ideas would be better. Especially if you include sea ice area trends in Figure 2, then you could start with sea ice change, then wind speed change, then stress change (which is a combo of both effects), and thus build to the analysis of "which one was it?". Thank you for this suggestion. We have revised accordingly. We maintained the original order of ideas and introduced the discussion on sea ice area later. We believe that starting with the discrepancy between wind and stress changes provides an interesting starting point and good motivation for exploring the relative roles of ice and wind, as discussed later.

- L138: As with the previous comment, this is awkward phrasing. Rephrased accordingly to: "Similar to surface stress, winds are generally weaker during summer "

- L185–187 and Fig 2: In fact, the Winter stress trend matches what might be predicted by the wind speed change pretty well (Winter wind speed trend +18%, and quadratic drag gives $1.18^2=1.39$ or a +39% stress increase, consistent with Winter stress trend +38% in Figure 2a). It's the Summer and Fall trends that are weird: why is stress increasing more slowly than wind speed during those seasons? Please provide some commentary on this (including the expectations based only on wind speed changes). Thank you for raising this important question. We acknowledge that the initial explanation in this section was unclear, and the total increase by the end of the century was confusing when mixed with the trends, which is our focus. When considering trends from 2020-2060, wind speed trends are largest in fall, while surface stress trends are largest in winter. Specifically, the winter change is not quadratic. The winter wind speed trend is +1.2% per decade. According to the quadratic drag law, this would result in \$1.012^2 = 1.024\$ or a +2.4% stress increase, which does not match the observed +5.1% per decade increase in stress. For fall, the trend is also not quadratic: \$1.0216^2 = 1.044\$, or a +4.4% stress increase per decade, compared to the simulated +2.2% increase. Therefore, in winter, the stress response is stronger than quadratic, whereas in summer and fall, the stress response is weaker than quadratic. This

simple calculation is however not exactly accurate. For a proper comparison of the expected stress change based on changing wind, one would have to calculate the quadratic change for every grid cell, because mathematically, the average of the quadratic changes is not necessarily the same as the quadratic changes of the average.

To clarify that our focus is on trends and not the full period (which plateaus), we have changed the values in Figure 2 to represent the trends, rather than the change to the end of the century. The text has also been rewritten to ensure that the focus is on the trends, not the difference between the beginning and the end of the century. Additionally, we now present trend values as percentages to facilitate easy comparison between wind and stress trends. We agree that the lower stress response compared to wind speed in summer and fall is intriguing. We have added a brief discussion on this but continue to focus on the winter response, which is stronger than the wind change and the main focus of our paper. As mentioned, we suspect that the slower increase in stress during fall and summer and the flattening of the stress curves relates to the decline in sea ice areas to sub-optimal conditions for momentum transfer (Martin et al., 2016). The “plateauing” of the time series (Fig 2) is partially an artifact of the multi-model averaging, as individual models show a less coherent time evolution (Fig 3). The main result is a consistent response where, in fall, trends in wind speed are larger than surface stress trends, and in winter, surface stress trends are larger than wind speed trends. Our analysis further shows that this response is explained by the interplay of several processes. In the case study for NorESM, the reduced dissipation of momentum to internal sea ice stress contributes significantly to the increased transfer of momentum into the ocean in winter.

- L205: Awkward phrasing — the ocean model is forced by τ_o (which is computed from τ_{ao} and τ_{io}). Agreed, this has been rephrased accordingly.

- L208: Introduce NorESM2-MM as a "case study" in the introduction, and somewhere explain how representative it is. Agreed. As detailed in “major comments” this has now been introduced early and an discussion on the representativeness has been included.

- L218–219: Be more specific here. Stress out of the atmosphere is usually thought to be greater over ice than open water, which contrasts with the statement in the manuscript. This suggests that internal stresses are also included in that statement, but it isn't clear. Agree, this is unclear. The entire section has been restructured for clarity, but this specific sentence has been changed to: “Reduced ice area enhances momentum transfer as, in NorESM2-MM (and all other models), open water stress (atmosphere-ocean) is larger than ice-ocean stress (detailed in Section 2.4).”

- L279–282: This statement could be made more clear. Is the ice weaker because it has a lower concentration (on average), and strength is a function of concentration (and thickness)? Or is internal stress contribution less because the lower concentration means that there is less ice available to contribute to internal stress (integrating over a smaller area)? Are the results presented sufficient to allow us to separate those effects? Agreed. Following previous comments, this whole paragraph and the paragraph above has been expanded and reworded for clarity.

- L290–293: While the increase in stress is robust across models, it's not clear that there is sufficient evidence to say that the decomposition is robust across models (it probably is, but the results don't necessarily prove it). Agreed. This has been rephrased to the following: “Considering the similarity in ocean surface stress trends and understanding the principle physics implemented, we believe it is likely that the described mechanism is the dominant cause for the robust winter ocean surface stress amplification and seasonal variations at play across all models.”

- L297–301: This feels more like content for the introduction rather than the results. Agreed. This has been moved to the introduction section.
- Figure 4: I don't think the 2055-2070 mean adds much to the figure and it isn't discussed in the text (side note: this is a great figure). Agreed. We have removed the 2055-2070 mean.
- L363–364: Which previous studies suggest a shift in the seasonality of Arctic surface stress? Also, what would be the impact of this shift? This was suggested by Martin et al., (2014) which is now cited here.
- Figure 5: Fonts in the legend are small/hard to read. This has been changed in the updated Figure 5 (now Figure 6).
- L421/§4.3 #7: Does normalization by τ_a in the "damping factor" lead to issues when τ_a is a small value? We have investigated this, and \$\tau_a\$ does not get so close to zero that this becomes an issue.
- Section 2.5: Much of this content feels like background that would be more appropriate in an introduction section instead of a results section (e.g., L451-459 and especially L499-513). It seems that more could be said in this section (especially about shifting patterns of ocean surface velocity or kinetic energy), especially given the reference to these changes in the manuscript's title. We acknowledge that this section may appear relatively brief compared to others. To clarify that the main focus of the paper is on changes in surface stress and not the spin-up, we have revised the title. While the ocean response is a consequential aspect, we argue its importance in the broader narrative, making it relevant to a wider audience. However, due to space constraints, we were unable to delve deeply into the details. We have expanded this section slightly to emphasize the ice-ocean governor effect.
- L460/L495: Kinetic energy is not an appropriate proxy for turbulent mixing, which will depend on a balance of both kinetic and potential energy. (It is still appropriate to discuss kinetic energy in the context of mixing) This is a very good point and we fully agree. We do not equate KE with (turbulent) mixing which also depends on PE, aka stratification. We have rewritten this to: "For the other CMIP6 models, total kinetic energy (Methods) works as a proxy for the effect that the increased momentum input may have on vertical mixing, and exhibits a consistent increase across the model suite (Fig. S3)."
- L499-522: It isn't convincing why the Beaufort Gyre in particular is singled out for evaluation when there are a range of climatically important areas that could equally be considered. The sensitivity of some of the results (e.g., FWC) to other, likely more important factors (as described in L516–518), makes these results feel fairly contrived. If the authors plan to keep an evaluation of the Beaufort Gyre in future revisions, they should frame the analysis in terms of the "ice-ocean governor" effect (e.g., Meneghello et al., 2018) that they described in the introduction (I suspect that was the motivation in this evaluation, but it isn't linked back to those ideas). Thank you for this very good suggestion. We have incorporated a few additional sentences in this section to further connect it with the concept of the ice-ocean governor, as recommended.
- L547: Variability in bottom roughness should also cite Brenner et al., 2021 and Cole et al., 2017 (as already cited elsewhere in the manuscript). We agree that these references should be included here as well, and have done so.
- L647: "All trends presented are statistically significant unless otherwise stated" At what confidence level? All trends are statistically significant at the 95% confidence level. This detail has now been added

to L647.

- There are a few entries in the references with missing fields (indicated as "???"), and at least one with an error in the title (Castellani et al., 2014). Thank you for noticing this. The errors have been fixed by updating the Bibtex files.

REVIEWER COMMENTS

Reviewer #1 (Remarks to the Author):

The authors have done an excellent job revising this manuscript. They have addressed my concerns from the previous round and I am happy to recommend this manuscript for publication.

While reviewing I noticed a few minor typos and grammatical issues that should be addressed prior to publication.

Edward Doddridge

Line numbers refer to manuscript version with tracked changes.

I really appreciate the additional text in lines 65-73, and 373-381.

Lines 589-591: "The average multimodel mean" is tautological. The multimodel mean is, by definition, an average.

Line 647: "on internal stress is neither possible," Neither should be replaced by "not".

Reviewer #1 (Remarks on code availability):

I accessed the repository and downloaded the scripts. The files available match with what I was expecting, but I have not reviewed the code.

Reviewer #2 (Remarks to the Author):

Thank you to the authors for addressing my questions.

Reviewer #3 (Remarks to the Author):

I am happy to see the authors' thoughtful and thorough responses to my comments. The adjustments that they have made to the manuscript address most of my concerns and the revised

manuscript is much improved. I appreciate the additional analysis performed in response to my (and the other reviewers') comments and the time taken to verify parameterization schemes with different modelling centers. I have a few remaining thoughts and questions that I hope could be considered before publication.

- Fig 3 caption lists the labels "fall (top row)"/"winter (bottom row)" reversed relative to the actual figure.

- Section 2.2: I feel like an integral symbol instead of a summation symbol would better communicate what quantities represent. I have some confusion about the current notation in L348 and L360, where summations exist only on one side of the equation, and inconsistency about whether ice concentration is included. In general, while this revised section does better to communicate the results compared to the previous manuscript, it feels like it would be more impactful if it were simplified a bit.

- Section 2.4: I appreciate the discussion regarding drag uncertainty here and the expansion of the "damping effect" analysis. The current (new) subsection title doesn't do enough to highlight the very interesting analysis in this section. This section provides good evidence backing up claims in section 2.2 (that stress is increasing because of decreasing ice strength), and I feel they could be better linked. I am curious about the results of the damping effect analysis as they related to the previous version of the manuscript: in the present version, all damping effects are positive (energy loss); in the previous version, two of the models (CNRM-CM6-1 and IPSL-CM6A-LR) were negative. Did something change in the analysis to adjust the values for those models? The donut charts were a very clever display of the data in the previous manuscript version because the (consistent) clockwise response always indicated a decrease in damping regardless of whether the initial value was positive or negative.

- Section 2.5 and Fig 7: I appreciate the inclusion of ice thickness and speed evaluations. I think that this section could do more to link those changes to changing ice strength (links are mentioned, but the impact isn't very strong). In particular, the sea ice speed trends can be discussed in the context of wind changes. As with other effects, an increase in ice speed could be a response to changing winds or changing ice strength. The "wind factor"—the ratio of ice speed to wind speed—has been previously used as a proxy for ice rheology effects (Leppäranta & Omstedt, 1990 [doi:10.1034/j.1600-0870.1990.t01-2-00007.x]; Martini et al., 2014 [doi:10.1175/JPO-D-13-0160.1]; Dosser & Rainville, 2016 [doi:10.1175/JPO-D-15-0056.1]; Watkins et al., 2023 [10.1029/2023GL103558]; and others). Also see Olason & Notz (2014 [doi:10.1002/2014JC009897]) for further discussion about relationships between ice drift speed and ice strength, which may be of interest (though such analysis would not be within the current scope of the project).

Reviewer #3 (Remarks on code availability):

A README was not provided with the code, but the code is generally readable and straightforward for MATLAB users. However, it seems some code files load data files that are not included in the

source data (likely intermediate data products produced after some preprocessing of the source data).

Tromsø June 20, 2024

*Response to reviews of
"Future sea ice weakening amplifies wind-driven trends in surface stress
and Arctic Ocean spin-up"
submitted to Nature Communications (NCOMMS-24-00384-T)*

Thank you very much for the second round of constructive reviews of our submitted manuscript. We are pleased that the manuscript and our revisions were positively received. We have addressed all remaining comments and detailed our responses point-by-point in **blue font** below. Changes to the manuscript have been made accordingly, and we are happy to say that it has improved the manuscript. Additionally, a tracked-changes document is attached for your reference.

On behalf of all the authors,
Morven Muilwijk

Reviewer #1 (Remarks to the Author):

The authors have done an excellent job revising this manuscript. They have addressed my concerns from the previous round and I am happy to recommend this manuscript for publication. While reviewing I noticed a few minor typos and grammatical issues that should be addressed prior to publication.
Edward Doddridge

Thank you very much for this constructive review and for noticing these typos and grammatical issues. We are happy that our revisions were appreciated.

I really appreciate the additional text in lines 65-73, and 373-381. **Thank you for this.**

Lines 589-591: "The average multimodel mean" is tautological. The multimodel mean is, by definition, an average. **Very good point. "average" has been removed.**

Line 647: "on internal stress is neither possible," Neither should be replaced by "not". **"neither has been replaced with "not".**

Reviewer #1 (Remarks on code availability):

I accessed the repository and downloaded the scripts. The files available match with what I was expecting, but I have not reviewed the code.

Reviewer #2 (Remarks to the Author):

Thank you to the authors for addressing my questions.

It is our pleasure and thank you for taking the time to carefully review our paper. We are glad that the revisions were satisfactory.

Reviewer #3 (Remarks to the Author):

I am happy to see the authors' thoughtful and thorough responses to my comments. The adjustments that they have made to the manuscript address most of my concerns and the revised manuscript is much improved. I appreciate the additional analysis performed in response to my (and the other reviewers') comments and the time taken to verify parameterization schemes with different modelling centers. I have a few remaining thoughts and questions that I hope could be considered before publication.

Thank you for your positive feedback and for your remaining thoughts and questions. We are happy the adjustments addressed your concerns and will reply to your remaining questions below.

- Fig 3 caption lists the labels "fall (top row)"/"winter (bottom row)" reversed relative to the actual figure. Thank you for noticing this. Top and bottom row have now been switched in the caption.

- Section 2.2: I feel like an integral symbol instead of a summation symbol would better communicate what quantities represent. I have some confusion about the current notation in L348 and L360, where summations exist only on one side of the equation, and inconsistency about whether ice concentration is included. In general, while this revised section does better to communicate the results compared to the previous manuscript, it feels like it would be more impactful if it were simplified a bit.

Thank you for this suggestion. We agree that the integral notation more accurately reflects the continuous nature of the spatial domain over which the wind stress is integrated. In response to your suggestion, we have revised the manuscript to replace all summation symbols with integrals in the form $\int_{\text{Arctic}} \tau, dS$. In L348 we have included the sea ice concentration on the left side and the integral on the right side. In L360, the left side of the equation was not needed and has been removed. To simplify, we have excluded the equations that are previously defined and repeated in L328, L337, L341, L346 and L362. We believe removing the many equations has made the section more impactful, but we have chosen not to shorten it further as other reviewers were happy with the additional explanations added in the previous round.

- Section 2.4: I appreciate the discussion regarding drag uncertainty here and the expansion of the "damping effect" analysis. The current (new) subsection title doesn't do enough to highlight the very interesting analysis in this section. This section provides good evidence backing up claims in section 2.2 (that stress is increasing because of decreasing ice strength), and I feel they could be better linked. I am curious about the results of the damping effect analysis as they related to the previous version of the manuscript: in the present version, all damping effects are positive (energy loss); in the previous version, two of the models (CNRM-CM6-1 and IPSL-CM6A-LR) were negative. Did something change in the analysis to adjust the values for those models? The donut charts were a very clever display of the data in the previous manuscript version because the (consistent) clockwise response always indicated a decrease in damping regardless of whether the initial value was positive or negative.

Thank you for your comments. We are pleased that the new analysis is appreciated. To highlight the results in this section, we have changed the subsection title to "Reduced dampening effect". In the initial version, the results from CNRM-CM6-1 and IPSL were indeed different. While refining the analysis of the dampening effect and including all other models, we identified an error in the code. Specifically, the atmospheric variables in CNRM and IPSL used a different longitude grid than their ocean grid, resulting in

a mismatch that excluded a sector of the Arctic Ocean from the atmospheric analysis but not from the ocean analysis. Fortunately, this error has been corrected, and we are confident that the new analysis is accurate. There is now greater agreement among the models, which, as you point out, strengthens the results and supports the claims in Section 2.2. We considered changing the data display (donut charts) but believe it remains an effective way to present these results. The climate response is consistent (clockwise = decrease in damping), and the donut charts provide a clear visualization of the percentage (a part of the energy lost), which is less obvious and harder to compare in bar plots.

- Section 2.5 and Fig 7: I appreciate the inclusion of ice thickness and speed evaluations. I think that this section could do more to link those changes to changing ice strength (links are mentioned, but the impact isn't very strong). In particular, the sea ice speed trends can be discussed in the context of wind changes. As with other effects, an increase in ice speed could be a response to changing winds or changing ice strength. The "wind factor"—the ratio of ice speed to wind speed—has been previously used as a proxy for ice rheology effects (Leppäranta & Omstedt, 1990 [doi:10.1034/j.1600-0870.1990.t01-2-00007.x]; Martini et al., 2014 [doi:10.1175/JPO-D-13-0160.1]; Dosser & Rainville, 2016 [doi:10.1175/JPO-D-15-0056.1]; Wolaatkins et al., 2023 [10.1029/2023GL103558]; and others). Also see Olason & Notz (2014 [doi:10.1002/2014JC009897]) for further discussion about relationships between ice drift speed and ice strength, which may be of interest (though such analysis would not be within the current scope of the project).

Thank you for your good feedback on Section 2.5. We agree that the links to changing ice strength could be more clearly articulated. Consequently, we have added additional references to Section 2.2 to enhance these connections. We acknowledge that the "wind factor" would be an interesting diagnostic to compare in the CMIP6 models. However, this analysis is beyond the scope of our current paper, which is already long. Additionally, direct comparisons between early and later periods are challenging due to significant changes in sea ice area and distribution, which complicate consistent comparisons. To acknowledge this point, we have included the following statement in the manuscript: "As with other effects, the increase in ice speed could be a response to changing winds or changing ice strength. Previous studies have used the ratio of ice speed to wind speed as a proxy for ice rheology \citep[e.g.]{}{Lepparanta1990, olason2014drivers}, but such analysis is beyond the scope of this study. We hope these revisions address your concerns, and we appreciate your valuable suggestions.

Reviewer #3 (Remarks on code availability):

A README was not provided with the code, but the code is generally readable and straightforward for MATLAB users. However, it seems some code files load data files that are not included in the source data (likely intermediate data products produced after some preprocessing of the source data). We have now included a README file with the code. Most code processes raw CMIP6 data sourced from the ESGF server. Due to its size (multiple terabytes), this data is not included in the SourceData but is linked in the Data Availability Statement. This code reads raw data for each individual model (4D fields) that have been preprocessed to ensure consistent variable names and dimensions across models. The SourceData file includes the output (time series) generated by these scripts. Some code loads intermediate data products, which are spatial integrals of the raw data. These spatial integrals have now been included in the SourceData file.

REVIEWERS' COMMENTS

Reviewer #3 (Remarks to the Author):

I appreciate the authors' patience with my questions and comments throughout the review process. My concerns have now all been addressed, and I am happy to recommend the current version of the manuscript for publication.

Reviewer #3 (Remarks on code availability):

The code now contains relevant details to run, in the new README